# GDPO: Group reward-Decoupled Normalization Policy Optimization for Multi-reward RL Optimization

**Shih-Yang Liu** [1 2]  **Xin Dong** [1]  **Ximing Lu** [1]  **Shizhe Diao** [1]  **Peter Belcak** [1]  **Mingjie Liu** [1]  **Min-Hung Chen** [1]
**Hongxu Yin** [1]  **Yu-Chiang Frank Wang** [1]  **Kwang-Ting Cheng** [2]  **Yejin Choi** [1]  **Jan Kautz** [1]  **Pavlo Molchanov** [1]

## Abstract

Recent work has defaulted to apply Group Relative Policy Optimization (GRPO) under multi-reward setting without examining its suitability. In this paper, we demonstrate that directly applying GRPO to normalize distinct rollout reward combinations causes them to collapse into identical advantage values, reducing the resolution of the training signal and resulting in suboptimal convergence and, in some cases, early training failure. We then introduce **G**roup reward-**D**ecoupled Normalization **P**olicy **O**ptimization (**GDPO**), a new policy optimization method to resolve these issues by decoupling the normalization of individual rewards, more faithfully preserving their relative differences and enabling more accurate multi-reward optimization, along with substantially improved training stability. We compare GDPO with GRPO across three tasks: tool calling, math reasoning, and coding reasoning, evaluating both correctness metrics (accuracy, bug ratio) and constraint adherence metrics (format, length). Across all settings, GDPO consistently outperforms GRPO, demonstrating its effectiveness and generalizability for multi-reward reinforcement learning optimization. Code is available at
https://github.com/NVlabs/GDPO.

## 1. Introduction

As language models continue to advance in capability, expectations for their behavior have grown accordingly. Demand for models to not only provide accurate responses but also exhibit behaviors aligned with a wide range of human preferences across diverse scenarios has continued to increase. These preferences span efficiency (Liu et al., 2025d;

[1]NVIDIA [2]HKUST. Correspondence to: Shih-Yang Liu <shihyangl@nvidia.com, sliuau@connect.ust.hk>.

*Proceedings of the 43$^{rd}$ International Conference on Machine Learning*, Seoul, South Korea. PMLR 306, 2026. Copyright 2026 by the author(s).

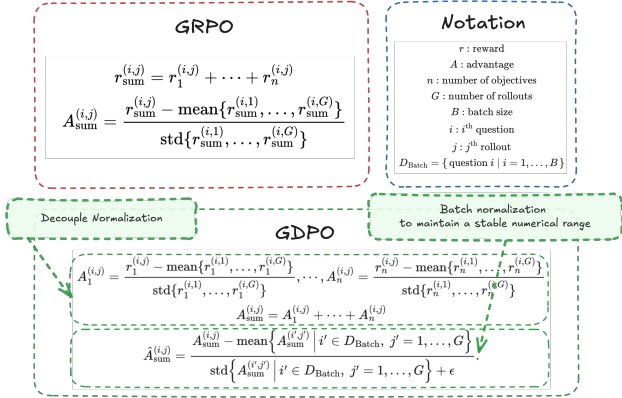

*Figure 1.* An overview of GDPO, which performs group-wise normalization per reward and then applies batch-wise advantage normalization to preserve a stable numerical range independent of reward count and improve update stability.

Team et al., 2025; Aggarwal & Welleck, 2025b), safety (Mu et al., 2024), response coherence and logic (Chen et al., 2025b; Liu et al., 2025a), gender biases (Zhang et al., 2025b) and many other objectives. Meeting such heterogeneous requirements within a single model is a challenging task.

Reinforcement learning (RL) has emerged as the de facto training pipeline for aligning large language models to fulfill such diverse human preferences. In particular, recent RL-based approaches have begun to incorporate multiple rewards into training, with each reward designed to capture different human preferences and collectively guide models toward human-favored behaviors. Despite this growing interest in multi-reward RL, recent work (Liu et al., 2025d; Aggarwal & Welleck, 2025b; Chen et al., 2025b) has largely focused on the reward design itself and often directly relied on applying Group Relative Policy Optimization (GRPO) directly for multi-reward RL optimization, often without examining whether GRPO is well-suited for optimizing combinations of heterogeneous rewards.

In this paper, we revisit the applicability of GRPO in multi-reward settings and show that directly applying GRPO to normalize different combinations of rollout rewards can cause them to collapse into identical advantage values,

which effectively limits the precision of the training signal, as illustrated in Fig. 2(a). This collapse removes important distinctions across reward dimensions and leads to inaccurate policy updates, suboptimal reward convergence, and, in many cases, early training failure.

To overcome these challenges, we propose **Group reward-Decoupled Normalization Policy Optimization (GDPO)** which decouples the group-wise normalization of each individual reward as illustrated in Fig. 1, to ensure that distinctions across different reward combinations are better preserved and more accurately reflect the relative differences in model responses. This leads to more precise multi-reward optimization and substantially improved training convergence. After this decoupled group-wise normalization, we apply batch-wise advantage normalization to ensure that the magnitude of advantage does not increase as the number of individual rewards increases.

We compare GDPO and GRPO across three tasks: tool calling, math reasoning, and code reasoning. These tasks cover a wide range of objectives, including tool-calling accuracy and format correctness, mathematical reasoning accuracy and adherence to reasoning-length constraints, and code pass rate and bug ratio. Across all tasks, GDPO converges better. For example, in Fig. 3, training Qwen2.5-1.5B-Instruct with GDPO attains both higher correctness and format compliance than GRPO on the tool-calling task. On challenging math tasks, GDPO consistently outperforms GRPO. For instance, training DeepSeek-R1-1.5B and Qwen3-4B-Instruct with GDPO yields up to 6.3% and 2.3% higher accuracy on AIME compared to GRPO, while keeping more responses short simultaneously.

The summary of our contributions are as follows:

- **Analysis of GRPO reward collapse.** We demonstrate that applying GRPO naively for multi-reward RL optimization can collapse distinct rollout reward combinations into identical advantage values, thereby diminishing the resolution of the learning signal.

- **Remediation of GRPO reward collapse.** We propose GDPO, which performs group-wise decoupled normalization of each reward separately to better preserve cross-reward distinctions and enable more accurate multi-reward optimization.

- We carry out extensive experiments on three tasks: tool calling, math reasoning, and code reasoning, and compare the effectiveness of GDPO on optimizing a wide range of rewards corresponding to accuracy, format correctness, length constraints, and code quality. In all settings, GDPO consistently outperforms GRPO, showing improved training convergence and stronger downstream performance that align more closely with a diverse set of preferences.

## 2. GRPO's Propensity for Reward Signal Collapse in Multi-Reward RL

Recent advancements such as Group Relative Policy Optimization (GRPO) (Guo et al., 2025) and its variants, including DAPO (Yu et al., 2025a) and Reinforce++-Baseline (Hu et al., 2025), have emerged as widely adopted reinforcement learning algorithms due to their efficiency and simplicity. In contrast to Proximal Policy Optimization (PPO) (Schulman et al., 2017), GRPO eliminates the need for a value model by leveraging group-relative advantage estimation for policy updates.

Currently, GRPO has been primarily employed for optimizing a single-objective reward, typically focusing on accuracy. However, as model capability continues to grow, recent works have increasingly sought to optimize multiple rewards, such as response length constraint and formatting quality, in addition to accuracy (Liu et al., 2025d; Qian et al., 2025; Aggarwal & Welleck, 2025b), to better align with human preferences. Existing approaches for multi-reward RL generally adopt a straightforward strategy: summing all reward components and applying GRPO directly.

Formally, for a given question–answer pair $(q_i, o_j)$, where the behavior policy $\pi_{\theta_{old}}$ samples a group of $G$ responses $\{o_j\}_{j=1}^{G}$, and assuming $n$ objectives, the aggregated reward for the $j$-th response is computed as the sum of each objective's reward:

$$r_{sum}^{(i,j)} = r_1^{(i,j)} + \cdots + r_n^{(i,j)} \tag{1}$$

The group-relative advantage for the $j$-th response is then obtained by normalizing the group-level aggregated rewards:

$$A_{sum}^{(i,j)} = \frac{r_{sum}^{(i,j)} - \text{mean}\{r_{sum}^{(i,1)}, \ldots, r_{sum}^{(i,G)}\}}{\text{std}\{r_{sum}^{(i,1)}, \ldots, r_{sum}^{(i,G)}\}} \tag{2}$$

The corresponding multi-reward GRPO optimization objective can then be expressed as

$$\mathcal{J}_{GRPO}(\theta) = \mathbb{E}_{(q_i, o_j) \sim D, \{o_j\}_{j=1}^{G} \sim \pi_{\theta_{old}}(\cdot|q)}$$

$$\left[ \frac{1}{G} \sum_{j=1}^{G} \frac{1}{|o_j|} \sum_{t=1}^{|o_j|} \min\left( s_{i,t}(\theta) A_{sum}^{(i,j)}, \right. \right. \tag{3}$$

$$\left. \left. \text{clip}\left( s_{i,t}(\theta), 1 - \epsilon, 1 + \epsilon \right) A_{sum}^{(i,j)} \right) \right]$$

We first revisit this common practice of applying GRPO for mulit-reward RL optimization and identify a previously overlooked issue, that is GRPO inherently compresses the reward signal, causing loss of information in the advantage estimates. To illustrate, we start with a simple training setting and then extend it to more general cases. Consider a scenario where we generate two rollouts for each question

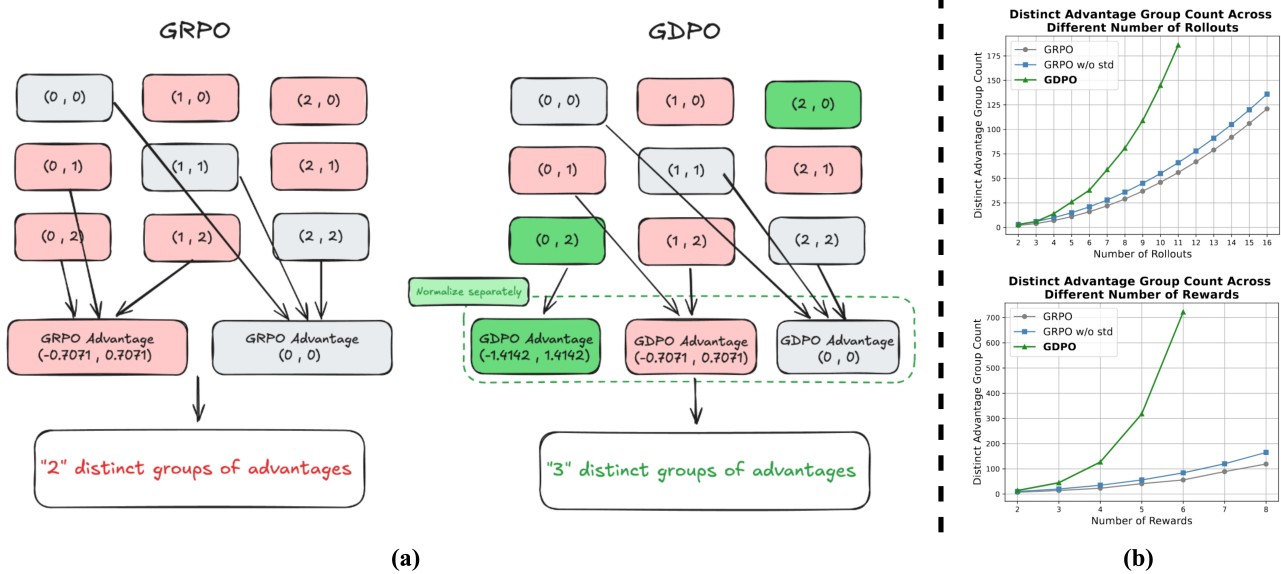

**(a)**                                   **(b)**

*Figure 2.* **(a):** Comparison of GRPO and GDPO advantage computation in a two-binary-reward, two-rollout example. GRPO maps different reward combinations into only two distinct advantage groups, whereas GDPO normalizes each reward independently and retains three distinct groups of advantage values. We skip the batch-wise normalization calculation step in GDPO here for simplicity since it does not change the number of distinct advantage groups. **(b):** Comparison of the number of distinct advantage groups produced by GRPO, GRPO without standard deviation normalization (GRPO w/o std), and GDPO. As the number of rollouts (up) or rewards (down) grows, GDPO consistently preserve a substantially larger number of distinct advantage groups compared to GRPO and GRPO w/o std. This results in advantage estimations that provide more expressive training signals.

for calculating the group-relative advantage and the task involves two binary reward $r_1, r_2 \in \{0, 1\}$. Consequently, the total reward for each rollout can take values from $\{0, 1, 2\}$.

As shown in Fig. 2(a), we enumerate all possible rollout reward combinations within a group, represented as (`rollout 1's total reward`, `rollout 2's total reward`) and the corresponding normalized advantages as (*rollout 1's advantage*, *rollout 2's advantage*). Despite having six distinct combinations when order is ignored, only two unique advantage groups emerge after applying group-wise reward normalization. Specifically, $(0, 1)$, $(0, 2)$, and $(1, 2)$ yield identical normalized advantages $A_{\text{sum}}$ of $(-0.7071, 0.7071)$, while $(0, 0)$, $(1, 1)$, and $(2, 2)$ all result in $(0, 0)$.

This demonstrates a fundamental limitation of GRPO's advantage calculation in multi-reward optimization which over-compresses the rich group-wise reward signal. Intuitively, $(0, 2)$ should produce a stronger learning signal than $(0, 1)$ because a total reward of 2 indicates simultaneous satisfaction of two rewards, whereas a reward of 1 corresponds to achieving only one. Thus, when the other one rollout only receives zero reward, $(0, 2)$ should yield a larger relative advantage than $(0, 1)$. This limitation can also introduce risks of training instability due to inaccurate advantage estimates. As shown in Fig. 4, the correctness reward score begins to decline after approximately 400 training steps when training

with GRPO, indicating a partial training collapse.

Recently, Dr.GRPO (Liu et al., 2025e) and DeepSeek-v3.2 (Liu et al., 2025a) adopt a variant of GRPO that removes the standard deviation normalization term from Eq. 2, such that $A_{\text{sum}}^{(i,j)} = r_{\text{sum}}^{(i,j)} - \text{mean}\{r_{\text{sum}}^{(i,1)}, \ldots, r_{\text{sum}}^{(i,G)}\}$. Despite these works introduce this modification to mitigate question-level difficulty bias, at first glance, this change also appears to address the issue we identify. Specifically, removing the standard deviation normalization mitigates the issue: $(0, 1)$ and $(0, 2)$ now yield distinct advantages of $(-0.5, 0.5)$ and $(-1.0, 1.0)$, respectively. However, when this setup is generalized to a larger number of rollouts while keeping the number of rewards fixed, as shown in Fig. 2(b), we observe that such fix only slightly increases the number of distinct advantage groups compared to GRPO. A similar trend can be observed under settings where the number of rollouts is fixed at four, but the number of rewards gradually increases. In this case, we also observe only modest improvements in the number of distinct advantage groups. We also empirically examine the effectiveness of removing the standard deviation normalization term in Section 4.1.1, and find that this modification does not lead to improved convergence or better downstream evaluation performance.

# 3. Method

## 3.1. Group reward-Decoupled normalization Policy Optimization

To overcome these challenges, we propose **Group reward-Decoupled normalization Policy Optimization (GDPO)**, a method designed to better maintain distinctions among different reward combinations and more accurately capture their relative differences in the final advantages. In contrast to GRPO, which applies group-wise normalization directly to the aggregated reward sum, GDPO decouples this process by performing group-wise normalization of each reward separately before aggregation. Concretely, rather than summing all $n$ rewards first (as in Eq. 1) and then applying group-wise normalization to obtain $A_{\text{sum}}$ (Eq. 2), GDPO computes the normalized advantage for each reward for the $j^{\text{th}}$ rollout of the $i^{\text{th}}$ question as:

$$A_1^{(i,j)} = \frac{r_1^{(i,j)} - \text{mean}\{r_1^{(i,1)}, \ldots, r_1^{(i,G)}\}}{\text{std}\{r_1^{(i,1)}, \ldots, r_1^{(i,G)}\}}, \ldots, A_{n-1}^{(i,j)}, A_n^{(i,j)} \tag{4}$$

The overall advantage used for policy updates is then obtained by first summing the normalized advantages across all objectives:

$$A_{\text{sum}}^{(i,j)} = A_1^{(i,j)} + \cdots + A_n^{(i,j)} \tag{5}$$

$$\hat{A}_{\text{sum}}^{(i,j)} = \frac{A_{\text{sum}}^{(i,j)} - \text{mean}\left\{A_{\text{sum}}^{(i',j')} \mid i' \in D_{\text{Batch}}, \ j' = 1, \ldots, G\right\}}{\text{std}\left\{A_{\text{sum}}^{(i',j')} \mid i' \in D_{\text{Batch}}, \ j' = 1, \ldots, G\right\} + \epsilon} \tag{6}$$

then applying batch-wise advantages normalization to the sum of the multi-reward advantages, which ensures that the numerical scale of the final advantage $\hat{A}_{\text{sum}}^{(i,j)}$ remains stable and does not grow as additional rewards are introduced. Empirically, we also find that this normalization step improves training stability, as shown in Appendix A, where removing batch-wise normalization occasionally leads to convergence failures. We also provide an example implementation of GDPO based on verl (Sheng et al., 2024) in Appendix B.

By separating the normalization of each reward, GDPO alleviates the information-loss problem present in GRPO's advantage estimation, as illustrated in Fig. 2(a). Note that since the batch-wise normalization step in GDPO does not alter the number of distinct advantage groups, we omit it here for clarity. From the figure, we can see that when adopting GRPO, distinct reward combinations, such as $(0, 2)$ and $(0, 1)$, lead to identical normalized advantages, masking the subtle distinctions between them. In contrast, GDPO retains these fine-grained differences by assigning distinct advantage values to each combination, for example, the reward combination of $(0, 1)$ after GDPO normalization becomes $(-0.7071, 0.7071)$ and $(0, 2)$ becomes $(-1.4142, 1.4142)$, which more appropriately reflects that $(0, 2)$ should yield a stronger learning signal than

$(0, 1)$. Similarly, when extending the number of rollouts to three, GRPO would assign advantage values of $(0, 0, 0)$ to $(1, 1, 1)$. However, $(1, 1, 1)$ may arise from heterogeneous reward partitions such as $r_1 = (1, 1, 0)$ or $r_2 = (0, 0, 1)$, for which GDPO would yield non-zero advantages, thereby preserving meaningful differences across reward dimension.

We further quantify the effectiveness of GDPO by comparing the number of distinct advantage groups across GDPO, GRPO, and GRPO w/o std under two experimental settings as shown in Fig. 2(b). In the two-reward scenario with a varying number of rollouts, GDPO consistently produces a significantly higher count of distinct advantage groups, with the gap widening as the number of rollouts increases. On the other hand, when fixing the number of rollouts to four and increasing the number of rewards, a similar pattern emerges, where GDPO exhibits progressively larger advantage granularity as the objective count grows. This demonstrate that the proposed decoupled normalization approach effectively increases the number of distinct advantage groups across all the RL settings and enables more precise advantage estimation. In addition to these theoretical improvements, we observe that using GDPO consistently yields a more stable training curve and improved convergence. For instance, GDPO achieves better convergence on both the format reward and the correctness reward in the tool-calling task, as shown in Fig. 3. GDPO also eliminates the training collapse issue observed with GRPO in the math reasoning task, as shown in Fig. 4, where the model trained with GDPO continues to improve the correctness reward score throughout training. Additional empirical results in Section 4 further confirm GDPO's ability to achieve stronger alignment with the target preferences across a wide range of downstream tasks.

# 4. Experiments

We begin by evaluating the effectiveness of GDPO compared with GRPO on the tool-calling task (Sec. 4.1), which involves optimizing two rewards: tool-calling correctness and format compliance. We then present an ablation study that examines the training convergence and downstream performance of GRPO with and without standard deviation normalization. Next, we compare GDPO and GRPO on a math reasoning task that optimizes two implicitly competing rewards, accuracy and length constraint (Sec. 4.2). We further conduct extensive analyses of the impact of incorporating different reward weights and modifying reward functions to better reflect varying priorities in human preferences, especially when rewards differ substantially in difficulty. Finally, we extend the number of optimized rewards to three and compare GRPO and GDPO on coding reasoning (Sec. 4.3), jointly optimizing code-generation accuracy, adherence to length constraints, and bug ratio, further demonstrating that

GDPO generalizes effectively to settings with three reward objectives.

## 4.1. Tool Calling

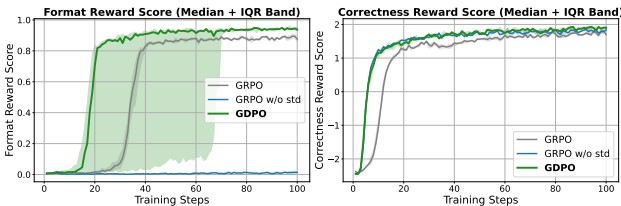

*Figure 3.* Median and IQR reward curves across five runs of Qwen2.5-1.5B on the tool-calling task for GDPO, GRPO, and GRPO w/o std.

We compare GDPO with GRPO on the tool calling task following the setup of ToolRL (Qian et al., 2025)[1]. Specifically, the model is trained to learn how to incorporate external tools into the reasoning trajectory to solve a user task following the output format shown in Appendix C, where the reasoning steps must be enclosed in `<think></think>`, the tool calls must appear within `<tool_call></tool_call>`, and the model's final answer must be placed inside `<response></response>`. We adopt the same training set as ToolRL for RL training, which consists of 2k samples from ToolACE (Liu et al., 2024), 1k samples from Hammar (Lin et al., 2024), and 1k samples from xLAM (Zhang et al., 2025a). Each training instance contains a question and its corresponding ground-truth tool calls. The training involves two rewards: **(i) a format reward** $\mathcal{R}_{\text{format}} \in \{0, 1\}$ that checks whether the model output satisfies the required structure and contains all necessary fields in the correct order, and **(ii) a correctness reward** $\mathcal{R}_{\text{correct}} \in [-3, 3]$ which evaluates the model-generated tool calls against the ground-truth calls using three metrics: tool name matching, parameter name matching, and parameter content matching.

A full description of the reward formulation is provided in Appendix D. We train Qwen-2.5-Instruct (1.5B and 3B) (Qwen et al., 2025) with GRPO and GDPO using verl (Sheng et al., 2024) for 100 steps, following the original hyperparameter settings from ToolRL's GRPO recipe. We use four rollouts per training question, a batch size of 512, and a maximum response length of 1024. The complete hyperparameter configuration is listed in Appendix E.

We evaluate the trained models on the Berkeley Function Call Leaderboard (BFCL-v3) (Patil et al.), a comprehensive benchmark covering a broad range of challenges, including single-step reasoning, multi-step tool use, real-time execution, irrelevant tool rejection, simultaneous multi-tool

---

[1]https://github.com/qiancheng0/ToolRL

selection, and multi-tool execution. We finetune the models with GRPO and GDPO across five runs and report the average accuracy and average format correctness on BFCL-v3 in Table 1. We additionally plot the median training curves with interquartile ranges for both methods over the five runs in Fig. 3.

From the training curves, we observe that GDPO consistently converges to higher values on both the format and correctness reward score across all runs. Although GDPO exhibits larger variance in the number of steps required to converge on format reward, it ultimately attains better format compliance than both GRPO. For the correctness reward, GDPO shows faster early-stage improvement and reaches a higher reward score than the GRPO baselines toward later stages, demonstrating the effectiveness of GDPO on providing more accurate advantage estimation that leads to better optimization.

*Table 1.* Comparison of GDPO and GRPO-trained Qwen2.5-Instruct-1.5B/3B models on tool-calling accuracy and format correctness. The reported results are the averages across five runs.

| | Live Acc ↑ | Multi Turn Acc ↑ | Non-Live Acc ↑ | Avg Acc ↑ | Correct Format ↑ |
|---|---|---|---|---|---|
| 1.5B | 37.89% | 0.12% | 15.63% | 17.88% | 4.74% |
| GRPO | 50.63% | 2.04% | 37.87% | 30.18% | 76.33% |
| GDPO | **55.36%** | **2.50%** | **40.58%** | **32.81%** | **80.66%** |
| 3B | 63.57% | 1.38% | 30.75% | 31.90% | 58.37% |
| GRPO | 69.23% | 3.14% | 45.24% | 39.20% | 81.64% |
| GDPO | **71.22%** | **4.59%** | **46.79%** | **40.87%** | **82.23%** |

In the BFCL-v3 evaluation shown in Table. 1, GDPO also consistently improves the average tool calling accuracy and format correctness over the GRPO-trained counterparts. For training Qwen2.5-Instruct-1.5B, GDPO achieves almost 5% and 3% improvement on Live/non-Live tasks and gains roughly 2.7% improvement on the overall average accuracy and more than 4% in correct format ratio compared with GRPO. Similar improvements are observed for the 3B model, where GDPO continues to outperform GRPO across all the sub-tasks, achieving up to 2% accuracy improvement and delivers a better format compliance ratio.

### 4.1.1. DOES REMOVING THE STANDARD DEVIATION NORMALIZATION TERM IN GRPO PROVIDE ANY BENEFIT?

*Table 2.* Comparison of GRPO, GRPO w/o std, and GDPO-trained Qwen2.5-Instruct-1.5B/3B models on tool-calling accuracy and format correctness. The reported results are the average across five runs.

| | Live Acc ↑ | Multi Turn Acc ↑ | Non-Live Acc ↑ | Avg Acc ↑ | Correct Format ↑ |
|---|---|---|---|---|---|
| 1.5B | 37.89% | 0.12% | 15.63% | 17.88% | 4.74% |
| GRPO | 50.63% | 2.04% | 37.87% | 30.18% | 76.33% |
| GRPO w/o std | 47.19% | 1.47% | 39.11% | 29.26% | 0% |
| GDPO | **55.36%** | **2.50%** | **40.58%** | **32.81%** | **80.66%** |

Recall from Fig. 2(b) that removing the standard deviation normalization term in GRPO (denoted GRPO w/o std)

slightly increases the number of distinct advantage groups. In this section, we empirically examine the effectiveness of this modification. Following the previous experiments, we run GRPO w/o std five times and report the average accuracy and average format correctness ratio on BFCL-v3.

In the reward training curves shown in Fig. 3, we observe that although GRPO w/o std converges to a correctness reward that is similar to GDPO and higher than standard GRPO, it fails to improve the format reward entirely. This failure results in a correct format ratio of 0% on BFCL-v3 as shown in Table. 2, indicating that the model does not learn the required output structure. These also show that simply removing the standard deviation normalization term in order to increase advantage diversity can introduce instability into training, which may ultimately prevent successful convergence in multi-reward reinforcement learning.

### 4.1.2. COMPATIBILITY WITH DIFFERENT POLICY OPTIMIZATION OBJECTIVES

In this section, we show that GDPO is complementary to the choice of policy optimization objective. As an advantage estimation method, GDPO can be seamlessly integrated into existing policy optimization frameworks, including GRPO (Guo et al., 2025), DAPO (Yu et al., 2025a), and CISPO (Chen et al., 2025a), by simply replacing the underlying advantage estimator.

For instance, consider the DAPO policy optimization objective:

$$\mathcal{J}_{\text{DAPO}}(\theta) = \mathbb{E}_{(q,a)\sim\mathcal{D},\{o_i\}_{i=1}^G\sim\pi_{\theta_{\text{old}}}(\cdot|q)}$$

$$\left[ \frac{1}{\sum_{i=1}^G |o_i|} \sum_{i=1}^G \sum_{t=1}^{|o_i|} \min\left( r_{i,t}(\theta)\,\hat{A}_{i,t}, \right.\right.$$

$$\left.\left. \text{clip}\left( r_{i,t}(\theta), 1 - \epsilon_{\text{low}}, 1 + \epsilon_{\text{high}} \right)\hat{A}_{i,t} \right) \right] \quad (7)$$

where $\hat{A}_{i,t}$ can be instantiated using either the GRPO advantage estimator (Eq. 3) or the proposed GDPO advantage estimator (Eq. 5). Likewise, GDPO can also be incorporated into the CISPO objective by substituting its original advantage estimation component.

To examine the generalizability of GDPO, we integrate GDPO into both DAPO and CISPO on the tool-calling benchmark, and compare the results against their counterparts using the original GRPO advantage estimation.

The results in Table. 3 show that replacing the original advantage estimator with GDPO consistently improves multi-reward performance under both DAPO and CISPO objectives. These findings suggest that GDPO is not coupled to any specific policy optimization objective, but instead serves

*Table 3.* Comparison of different policy optimization objectives on Qwen2.5-1.5B-Instruct for tool-calling performance and format correctness.

| | Live Acc ↑ | Multi Turn Acc ↑ | Non-Live Acc ↑ | Avg Acc ↑ | Correct Format ↑ |
|---|---|---|---|---|---|
| 1.5B | 37.89% | 0.12% | 15.63% | 17.88% | 4.74% |
| DAPO | 51.47% | 2.11% | 37.33% | 30.30% | 75.19% |
| DAPO w/ GDPO | 54.39% | 2.35% | 39.84% | 32.19% | 81.24% |
| CISPO | 46.43% | 1.87% | 35.16% | 27.82% | 77.48% |
| CISPO w/ GDPO | 52.41% | 2.08% | 37.72% | 30.74% | 79.15% |

as a broadly applicable enhancement for a wide range of policy optimization methods.

### 4.2. Mathematical Reasoning

We consider a mathematical reasoning task that optimizes two implicitly competing rewards: accuracy and adherence to a length constraint. The goal is to improve model performance on challenging mathematical problems while keeping the generated output within a predefined response length to encourage efficient problem solving. We train DeepSeek-R1-1.5B, DeepSeek-R1-7B (Guo et al., 2025), and Qwen3-4B-Instruct (Yang et al., 2025) using GRPO and GDPO on the DeepScaleR-Preview dataset (Luo et al., 2025b) for 500 steps, which contains 40k competition-level math problems. Training is performed using verl (Sheng et al., 2024), and we follow the original DeepSeek-R1 prompt format (Guo et al., 2025). Following the DLER setup (Liu et al., 2025c), we incorporate dynamic sampling, higher clipping thresholds, and the token-mean loss from DAPO (Yu et al., 2025a), and use 16 rollouts, a batch size of 512, and a maximum response length of 8000 tokens. The full set of hyperparameters is provided in Appendix F. For simplicity, throughout the following discussion, we use GRPO to refer to the combination of GRPO advantage estimation and the DAPO policy optimization objective, while GDPO refers to the combination of GDPO advantage estimation and the DAPO policy optimization objective.

The training uses two rewards: **(i) a length reward** $\mathcal{R}_{\text{length}} \in \{0, 1\}$ checks whether the model's output remains within the target length $l$, which is set to 4000 tokens for all remaining experiments, and **(ii) a correctness reward** $\mathcal{R}_{\text{correct}} \in \{0, 1\}$ indicates whether the final answer extracted from the model's response matches the ground truth.

We compare the GRPO and GDPO-trained model on AIME-24 (MAA, 2024a), AMC (AMC 2022 and AMC 2023) (MAA, 2024b), MATH (Hendrycks et al., 2021b), Minerva (Lewkowycz et al., 2022) and Olympiad Bench (He et al., 2024). All evaluations are conducted using vLLM as the inference backend with a sampling temperature of 0.6, $top_p = 0.95$, and a maximum response length of 32k tokens. For each evaluation question, we generate 16 samples and report the average pass@1 score and the average length-exceeding ratio, denoted Exceed, which measures the

percentage of model responses that exceed the predefined length limit of 4000 tokens.

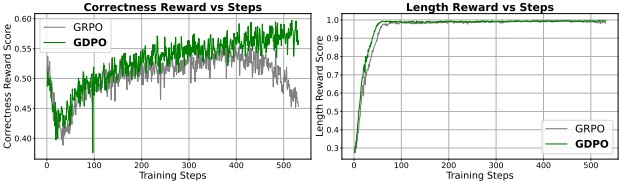

*Figure 4.* Training curves of GRPO and GDPO on DeepSeek-R1-1.5B across correctness reward and length reward.

From the training curves of GRPO and GDPO on DeepSeek-R1-1.5B as shown in Fig. 4, we first observe that the model tends to maximize the easier reward regardless of the optimization method. In this case, the length reward is easier to optimize, and both GRPO and GDPO reach a full length score within roughly the first 100 training steps. We also see that this rapid rise in the length reward coincides with an early drop in the correctness reward, which indicates that the two rewards are competing. During the initial phase of training, the model prioritizes satisfying the length constraint, often at the expense of the more challenging correctness objective. However, from the correctness reward trajectories, we observe that GDPO recovers the correctness reward more effectively than GRPO, achieving higher correctness scores at comparable training steps. We also see that GRPO training starts to destabilize after 400 steps with the correctness rewards score gradually decreasing while GDPO continue to improve the correctness score. Moreover, although both GDPO and GRPO maintain nearly perfect length scores throughout training, we also record the maximum response length within each training batch to assess how well the models satisfy the length constraint under more extreme cases in Fig.8. The results show that, despite achieving almost a full length reward, the maximum response length for GRPO begins to increase sharply after roughly 400 training steps, while the maximum response length for GDPO continues to decrease. Similar observation can be seen on the training curves on DeepSeek-R1-7B and Qwen3-4B-Instruct as shown in Fig 9 and Fig 10 in appendix where we can see that GDPO consistently provide better alignment to the length constraint. This contrast further illustrates the effectiveness of GDPO in multi-reward optimization compared with GRPO.

In addition, the benchmark results in Table. 4 show that the GDPO-trained models not only achieve substantial improvements in reasoning efficiency over the original models, with up to a 80% reduction in length-exceeding ratios on AIME, but also deliver higher accuracy on the majority of the tasks. Moreover, GDPO generally outperforms GRPO on both the accuracy and length constraint objectives. For the DeepSeek-R1-1.5B, GDPO outperforms GRPO across all benchmarks,

*Table 4.* Comparison of GRPO and GDPO-trained DeepSeek-R1-1.5B and DeepSeek-R1-7B models on Pass@1 accuracy and the proportion of responses exceeding the length constraint across mathematical reasoning benchmarks. See Table. 8 in Appendix for the results of Qwen3-4B-Instruct.

|  |  | DeepSeek-R1-1.5B | | | DeepSeek-R1-7B | | |
|---|---|---|---|---|---|---|---|
|  |  | - | GRPO | GDPO | - | GRPO | GDPO |
| MATH | Acc ↑ | 84.3% | 83.6% | **86.2%** | 93.6% | **94.1%** | 93.9% |
|  | Exceed ↓ | 35.0% | 1.5% | **0.8%** | 26.0% | 0.5% | **0.1%** |
| AIME | Acc ↑ | 29.8% | 23.1% | **29.4%** | 55.4% | 50.2% | **53.1%** |
|  | Exceed ↓ | 91.5% | 10.8% | **6.5%** | 85.6% | 2.1% | **0.2%** |
| AMC | Acc ↑ | 62.0% | 64.5% | **69.0%** | 82.9% | 83.8% | **84.0%** |
|  | Exceed ↓ | 67.5% | 3.2% | **2.3%** | 57.2% | 0.6% | **0.3%** |
| Minerva | Acc ↑ | 38.4% | 43.5% | **44.0%** | 49.8% | 53.2% | **53.8%** |
|  | Exceed ↓ | 51.4% | 1.7% | **0.3%** | 41.8% | 0.2% | **0.1%** |
| Olympiad | Acc ↑ | 44.1% | 44.3% | **46.6%** | 58.2% | **60.2%** | 59.7% |
|  | Exceed ↓ | 70.1% | 2.6% | **1.9%** | 60.6% | 1.1% | **0.4%** |

achieving accuracy improvements of 2.6%/6.7%/2.3% on MATH, AIME and Olympiad, respectively, while also reducing the length exceed ratios across all the tasks. A similar trend holds for DeepSeek-R1-7B, where GDPO achieves stronger accuracy–efficiency trade-offs. The gains are particularly notable on the more challenging AIME benchmark, with GDPO improving accuracy by nearly 3% while reducing the length-exceeding rate to 0.2%, compared with 2.1% under GRPO. Together, these results show that GDPO not only improves reasoning accuracy across a range of mathematical tasks but also adheres to the length constraint more effectively, underscoring its advantage in multi-reward optimization.

### 4.2.1. IMPACT ANALYSIS OF DIFFERENT REWARD PRIORITY VARIATION CONFIGURATIONS

So far, we have implicitly assumed equal importance across objectives. In practice, objectives often differ in priority, and a common approach is to encode such preferences via weighted reward aggregation, $r_{\text{sum}} = \sum_i w_i r_i$, or, in GDPO, by weighting normalized advantages: $A_{\text{sum}}^{(i,j)} = w_1 A_1^{(i,j)} + \cdots + w_n A_n^{(i,j)}$. We empirically study the effectiveness of this approach by fixing the correctness reward weight $w_{\text{correct}} = 1$ and varying the length reward weight $w_{\text{length}} \in \{0.25, 0.5, 0.75, 1.0\}$. As shown in Fig. 5, moderate reductions in $w_{\text{length}}$ have inconsistent effect on length violations for either GRPO or GDPO, and the relationship between $w_{\text{length}}$ and model behavior is often non-monotonic. Only when $w_{\text{length}}$ becomes sufficiently small (1.0 to 0.25) do we observe consistent relaxation of the length constraint, indicating that reward weighting alone does not reliably induce the intended prioritization. See Appendix Fig. 11 and Table. 10 for the full results across all tasks.

To address this limitation, following (Liu et al., 2025d;c),

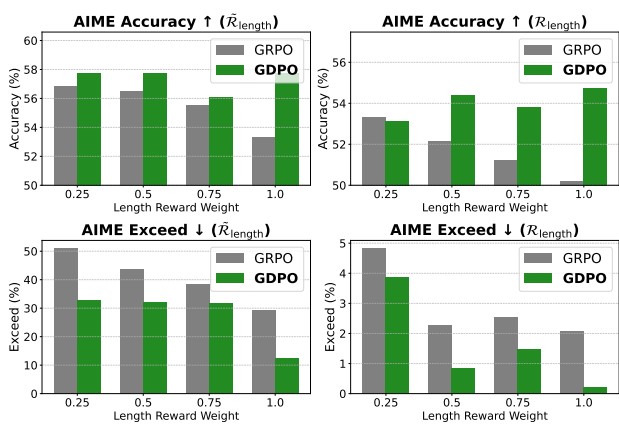

*Figure 5.* Average accuracy and exceed-length ratios for GRPO/GDPO-trained DeepSeek-R1-7B models under varying length reward weights $\{1.0, 0.75, 0.5, 0.25\}$, with and without the conditioned length reward $\tilde{\mathcal{R}}_{\text{length}}$, on AIME. See Fig. 11 in Appendix for the results on MATH.

we replace the original length reward $\mathcal{R}_{\text{length}}$ with a conditioned length reward:

$$\tilde{\mathcal{R}}_{\text{length}} = \begin{cases} 1, & \text{if response length} \leq l \text{ and } \mathcal{R}_{\text{correct}} = 1 \\ 0, & \text{otherwise.} \end{cases}$$

This design ensures that the length reward is only granted when the model answers the question correctly, preventing the model from exploiting the easier objective early in training.

As illustrated in Fig. 6, $\tilde{\mathcal{R}}_{\text{length}}$ helps stabilize training by avoiding early aggressive maximization of the length reward at the expense of correctness reward. Moreover, once the difficulty imbalance is mitigated, varying the conditioned reward weight $\tilde{w}_{\text{length}}$ leads to more faithful reflection of fine-grained preference adjustments. That is decreasing $\tilde{w}_{\text{length}}$ steadily increases length violations across tasks, in contrast to the unstable trends observed with unconditioned rewards in Fig. 5.

From Table. 9, we can also see that GDPO maintains a better accuracy–length trade-off than GRPO under the new reward $\tilde{\mathcal{R}}_{\text{length}}$. For example, On AIME, GDPO improves accuracy by 4.4% while reducing length violations by 16.9% compared to GRPO. This superior accuracy–efficiency trade-off holds across all configurations; see Sec. J for a more complete discussion.

### 4.3. Coding Reasoning

We examine whether GDPO continues to outperform GRPO when optimizing more than two rewards on our coding reasoning task. Similar to the mathematical reasoning setup, the objective is to improve the model's coding performance

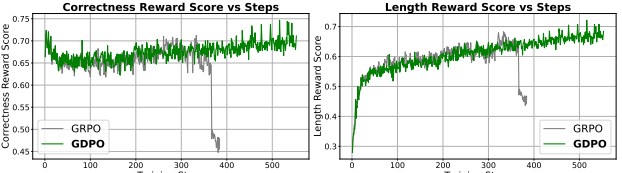

*Figure 6.* Training curves of GRPO and GDPO with the conditioned length reward $\tilde{\mathcal{R}}_{\text{length}}$ on DeepSeek-R1-7B across correctness reward, length reward.

*Table 5.* Comparison of GRPO and GDPO trained DeepSeek-R1-7B models on coding pass rate, length-exceeding rate, and bug ratio across coding reasoning benchmarks. Here, $\mathcal{R}_{\text{pass}} + \tilde{\mathcal{R}}_{\text{length}}$ refers to optimizing $\mathcal{R}_{\text{pass}}$ and $\tilde{\mathcal{R}}_{\text{length}}$, and $\mathcal{R}_{\text{pass}} + \tilde{\mathcal{R}}_{\text{length}} + \mathcal{R}_{\text{bug}}$ refers to optimizing $\mathcal{R}_{\text{pass}}$, $\tilde{\mathcal{R}}_{\text{length}}$, and $\mathcal{R}_{\text{bug}}$.

| | | | $\mathcal{R}_{\text{Pass}} + \tilde{\mathcal{R}}_{\text{length}}$ | | $\mathcal{R}_{\text{Pass}} + \tilde{\mathcal{R}}_{\text{length}} + \mathcal{R}_{\text{Bug}}$ | |
| | | - | GRPO$_{\text{2-obj}}$ | GDPO$_{\text{2-obj}}$ | GRPO$_{\text{3-obj}}$ | GDPO$_{\text{3-obj}}$ |
|---|---|---|---|---|---|---|
| | Pass ↑ | 28.1% | 67.2% | **68.3%** | **68.1%** | 67.8% |
| Apps | Exceed ↓ | 73.9% | 5.2% | **5.0%** | 11.2% | **8.5%** |
| | Bug ↓ | 32.9% | 25.0% | **23.5%** | 20.3% | **18.8%** |
| | Pass ↑ | 47.3% | 63.2% | **65.8%** | 65.6% | 65.6% |
| Codecontests | Exceed ↓ | 83.0% | **14.2%** | 14.3% | 19.3% | **15.8%** |
| | Bug ↓ | 29.7% | 14.1% | **13.2%** | 3.9% | **2.5%** |
| | Pass ↑ | 46.5% | 68.1% | **71.2%** | 69.5% | 69.4% |
| Codeforces | Exceed ↓ | 82.8% | **18.1%** | 18.4% | 16.9% | **13.6%** |
| | Bug ↓ | 27.8% | 7.0% | **5.6%** | 2.5% | **1.8%** |
| | Pass ↑ | 28.1% | 45.1% | **48.4%** | 44.4% | **45.1%** |
| Taco | Exceed ↓ | 78.0% | 11.8% | **10.8%** | 14.7% | **10.6%** |
| | Bug ↓ | 48.9% | 37.7% | **36.2%** | 30.0% | **28.0%** |

while constraining its output to a predefined target length. In addition, we introduce a third objective that encourages the model to generate bug-free code. We compare GDPO and GRPO by training DeepSeek-R1-7B on the Eurus-2-RL dataset (Cui et al., 2025), which contains 24k coding problems, each with multiple test cases. Training is conducted using the verl (Sheng et al., 2024) framework for 400 steps, and we adopt the same hyperparameter configuration used in the mathematical reasoning experiments. The training optimizes three rewards: **(i) a passrate reward** $\mathcal{R}_{\text{pass}} \in [0, 1]$ which measures the proportion of test cases passed by the generated code, **a conditioned length reward** $\tilde{\mathcal{R}}_{\text{length}} \in \{0, 1\}$ that checks whether the model's response remains within the target length $l$ and whether the generated code satisfies correctness requirements, and **(iii) a bug reward** $\mathcal{R}_{\text{bug}} \in \{0, 1\}$ indicates whether the generated code runs without runtime or compilation errors.

For evaluation, we assess the trained model on the validation set from PRIME (Cui et al., 2025), which includes Apps (Hendrycks et al., 2021a), CodeContests (Li et al., 2022), Codeforces[2], and Taco (Li et al., 2023). Following

---

[2]https://huggingface.co/datasets/MatrixStudio/Codeforces-

the same settings used for the mathematical reasoning evaluations, we use a sampling temperature of 0.6, a $top_p$ value of 0.95, and a maximum response length of 32k tokens. For each evaluation question, we generate 16 rollouts and report the average test case pass rate, the average length-exceeding ratio, and the average bug ratio, where the bug ratio measures the proportion of generated code that results in either a runtime error or a compilation error.

We compare GDPO and GRPO under two configurations: (1) a two-reward setting using $\mathcal{R}_{pass}$ and $\tilde{\mathcal{R}}_{length}$, and (2) a three-reward setting using $\mathcal{R}_{pass}$, $\tilde{\mathcal{R}}_{length}$, and $\mathcal{R}_{bug}$. We denote the two-reward and three-reward versions of GRPO as $GRPO_{2\text{-obj}}$ and $GRPO_{3\text{-obj}}$, and use the same notation for GDPO. As shown in Table. 5, $GDPO_{2\text{-obj}}$ improves pass rates across all the tasks compared with $GRPO_{2\text{-obj}}$, while maintaining a similar length-exceeding ratio. For example, $GDPO_{2\text{-obj}}$ improves the Codecontests pass rate by 2.6% while increasing the length-exceeding ratio by only 0.1%, and achieves a 3.3% pass rate gain together with a 1% reduction in length violations compared with $GRPO_{2\text{-obj}}$ on Taco. A similar pattern holds in the three-reward setting where $GDPO_{3\text{-obj}}$ achieves a substantially better balance across all objectives, maintaining similar pass rate to $GRPO_{3\text{-obj}}$ while also markedly reducing both the length-exceeding ratio and the bug ratio.

Overall, these results demonstrate that GDPO remains effective as the number of reward signals increases. It consistently achieves a more favorable trade-off across all objectives than GRPO in both the two-reward and three-reward configurations.

## 5. Related Work

**GRPO Variants**  Several extensions of Group Relative Policy Optimization (GRPO) (Shao et al., 2024) have been proposed to improve training stability, performance, and efficiency while preserving its core principles. For example, GSPO (Zheng et al., 2025) improves stability by performing sequence-level importance weighting and clipping, while DAPO (Yu et al., 2025b) enhances RL performance through higher clipping, dynamic sampling, token-level losses, and overlong reward shaping. To improve reasoning efficiency, GFPO (Shrivastava et al., 2025) mitigates length explosion via larger group sampling and response filtering, and DLER (Liu et al., 2025b) combines batch-wise normalization, higher clipping, dynamic sampling, and truncation penalties to achieve strong accuracy and efficiency trade-offs. Most related is BNPO (Xiao et al., 2025), which normalizes rewards via a dynamic Beta distribution for variance reduction and includes a secondary decomposition extension. In contrast, GDPO is motivated by a pre-

Python-Submissions

viously overlooked multi-reward collapse phenomenon in which distinct reward combinations collapse into identical advantages, and addresses it with a two-stage scheme of per-reward group-wise followed by post-aggregation batch-wise normalization, with both stages individually validated.

**Multi-Reward Reinforcement Learning**  A growing body of work studies reinforcement learning with multiple rewards, primarily to model diverse or competing human preferences. Examples include Safe RLHF (Dai et al., 2023), which balances helpfulness and harmlessness, RLPHF (Jang et al., 2023), which optimizes personalized preferences via policy merging, and ALARM (Lai et al., 2024), which uses hierarchical rewards to capture multiple response attributes. Recent LLM systems such as DeepSeek V3.2 (DeepSeek-AI et al., 2025) further combine outcome-based, length-based, and consistency rewards to enhance reasoning and agentic behavior.  Another major application of multi-reward RL focuses on improving reasoning efficiency through length-aware rewards, including normalized length penalties (Luo et al., 2025a; Arora & Zanette, 2025), group-relative conciseness objectives (Yi & Wang, 2025), explicit length control (Aggarwal & Welleck, 2025a), and adaptive accuracy–length trade-off strategies (Su & Cardie, 2025).

## 6. Conclusion

This work revisits the suitability of Group Relative Policy Optimization (GRPO) for multi-reward reinforcement learning and shows that directly optimizing summed rewards can collapse distinct reward combinations into identical advantages, leading to unstable training and degraded performance. To address this limitation, we introduce Group-wise Decoupled Policy Optimization (GDPO), a simple and effective modification to GRPO tailored for multi-reward reinforcement learning. GDPO performs normalization separately for each reward to preserve cross-reward differences, and it incorporates batch-wise advantage normalization to maintain a stable numerical range as additional rewards are included. These changes result in better convergence behavior and models that more faithfully preserve cross-reward distinctions. Across extensive experiments on tool calling, math reasoning, and coding tasks, GDPO consistently outperforms GRPO under varying numbers of rewards, model scales, and reward designs, demonstrating more stable convergence and better alignment with intended preference priorities.

## Impact Statement

This paper presents work whose goal is to advance the field of Machine Learning.  There are many potential societal consequences of our work, none which we feel must be specifically highlighted here.

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

# A. Training stability issue of GDPO without batch-wise advantage normalization

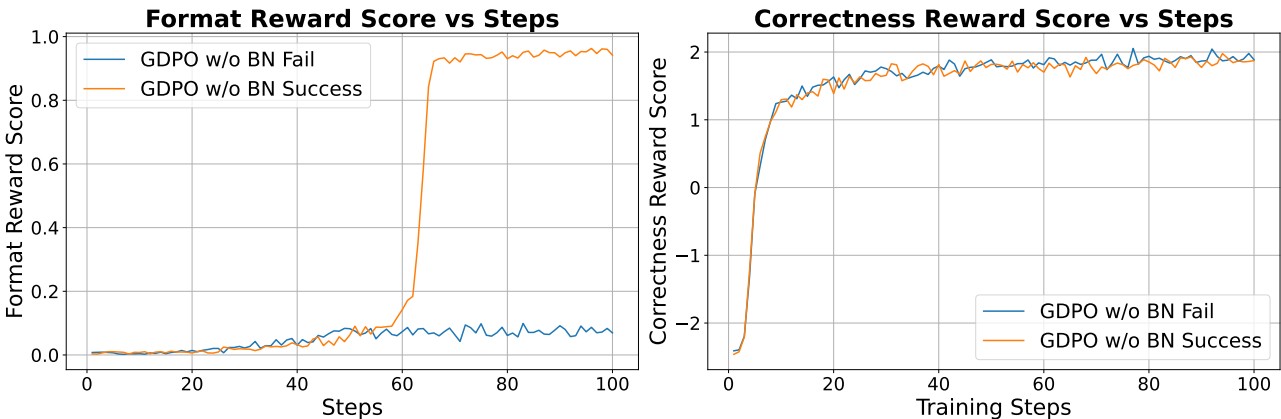

*Figure 7.* Training stability of GDPO with and without batch-wise advantage normalization. Runs without normalization occasionally fail to converge.

# B. GDPO vs GRPO Implementation based on verl

GDPO Implementation based on verl

```python
## Add the below implementation to ray_trainer.py
from verl.utils.torch_functional import masked_whiten
from verl.trainer.ppo import core_algos
...
if adv_estimator == 'gdpo':
        ## An example of using GDPO for two reward
        token_level_cor = data.batch['token_level_scores_cor']
        token_level_format = data.batch['token_level_scores_format']

        # shared variables
        index = data.non_tensor_batch['uid']
        responses = data.batch['responses']
        response_length = responses.size(-1)
        attention_mask = data.batch['attention_mask']
        response_mask = attention_mask[:, -response_length:]

        ## handle correctness first
        cor_norm_score, _ = core_algos.compute_grpo_outcome_advantage(token_level_rewards=token_level_cor,
                                                                      eos_mask=response_mask,
                                                                      index=index)

        ## handle format now
        format_norm_score, _ = core_algos.compute_grpo_outcome_advantage(token_level_rewards=token_level_format,
                                                                         eos_mask=response_mask,
                                                                         index=index)

        ## Sum up the decoupled-normalized advantage of each reward
        new_advantage = cor_norm_score + format_norm_score

        ## Apply Batch-normalization to ensure consistent numerical range
        advantages = masked_whiten(new_advantage, response_mask) * response_mask

        data.batch['advantages'] = advantages
        data.batch['returns'] = advantages
...
```

GRPO Implementation based on Verl

```python
from verl.trainer.ppo import core_algos
...
if adv_estimator == 'grpo':
        ## An example of using GRPO for two reward
        total_token_level_scores = data.batch['token_level_scores_cor'] + data.batch['token_level_scores_format']

        # shared variables
        index = data.non_tensor_batch['uid']
        responses = data.batch['responses']
        response_length = responses.size(-1)
        attention_mask = data.batch['attention_mask']
        response_mask = attention_mask[:, -response_length:]

        ## handle correctness first
        advantages, _ = core_algos.compute_grpo_outcome_advantage(token_level_rewards=total_token_level_scores,
                                                                  eos_mask=response_mask,
                                                                  index=index)

        data.batch['advantages'] = advantages
        data.batch['returns'] = advantages
...
```

## C. ToolRL Training Prompt Format

**System Prompt for ToolRL Training**

You are a helpful dialogue assistant capable of leveraging tool calls to solve user tasks and provide structured chat responses.

**Available Tools**
In your response, you can use the following tools:
`{ { Tool List } }`

**Steps for Each Turn**

1. **Think**: Recall relevant context and analyze the current user goal.

2. **Decide on Tool Usage**: If a tool is needed, specify the tool and its parameters.

3. **Respond Appropriately**: If a response is needed, generate one while maintaining consistency across user queries.

**Output Format**
`<think>` Your thoughts and reasoning `</think>`
`<tool_call>` {"name": "Tool name", "parameters": {"Parameter name": "Parameter content", " ... ...": " ... ..."}}
{"name": " ... ...", "parameters": {" ... ...": " ... ...", " ... ...": " ... ..."}}
...
`</tool_call>`
`<response>`AI's final response `</response>`

**Important Notes**

1. You must always include the `<think>` field to outline your reasoning. Provide at least one of `<tool_call>` or `<response>`. Decide whether to use `<tool_call>` (possibly multiple times), `<response>`, or both.

2. You can invoke multiple tool calls simultaneously in the `<tool_call>` fields. Each tool call should be a JSON object with a "name" field and a "parameters" field containing a dictionary of parameters. If no parameters are needed, leave the "parameters" field an empty dictionary.

3. Refer to the previous dialogue records in the history, including the user's queries, previous `<tool_call>`, `<response>`, and any tool feedback noted as `<obs>` (if exists).

**User Prompt for ToolRL Training**

**Dialogue History**

`<user>` { { Initial User Input } } `</user>`

`<think>`  Round 1 Model Thought  `</think>`
{ { Round 1 model output `<tool_call>` or `<response>` } }
`<obs>` Round 1 Observation `</obs>`

... ...

`<user>` { { User Input } } `</user>`

... ...

## D. Tool Calling Reward Functions

**Format Reward.** The format reward $\mathcal{R}_{\text{format}} \in \{0, 1\}$ checks whether the model output satisfies the required structure and contains all necessary fields in the correct order:

$$\mathcal{R}_{\text{format}} = \begin{cases} 1, & \text{if all required fields appear and are in the correct order,} \\ 0, & \text{otherwise.} \end{cases} \tag{8}$$

**Correctness Reward.** The correctness reward $\mathcal{R}_{\text{correct}} \in [-3, 3]$ evaluates the predicted tool calls $P = \{P_1, \ldots, P_m\}$ against the ground-truth calls $G = \{G_1, \ldots, G_n\}$. It consists of three components:

- **Tool Name Matching:**

$$r_{\text{name}} = \frac{|N_G \cap N_P|}{|N_G \cup N_P|} \in [0, 1],$$

  where $N_G$ and $N_P$ are the sets of tool names from ground-truth and predicted calls, respectively.

- **Parameter Name Matching:**

$$r_{\text{param}} = \sum_{G_j \in G} \frac{|\text{keys}(G_j) \cap \text{keys}(P_j)|}{|\text{keys}(G_j) \cup \text{keys}(P_j)|} \in [0, |G|],$$

  where $\text{keys}(G_j)$ and $\text{keys}(P_j)$ are the parameter names of the ground-truth and predicted calls.

- **Parameter Content Matching:**

$$r_{\text{value}} = \sum_{G_j \in G} \sum_{k \in \text{keys}(G_j)} \mathbf{1}[P_G[k] = P_P[k]] \in \left[0, \sum_{G_j \in G} |\text{keys}(G_j)|\right],$$

  where $P_G[k]$ and $P_P[k]$ are the parameter values for the ground-truth and predicted calls.

- **Total Match Score:**

$$r_{\text{match}} = r_{\text{name}} + r_{\text{param}} + r_{\text{value}} \in [0, S_{\text{max}}],$$

  where

$$S_{\text{max}} = 1 + |G| + \sum_{G_j \in G} |\text{keys}(G_j)|.$$

The final correctness reward is computed by finding the optimal matching between $P$ and $G$ to maximize the total match score:

$$\mathcal{R}_{\text{correct}} = 6 \cdot \frac{R_{\text{max}}}{S_{\text{max}}} - 3 \in [-3, 3].$$

where $R_{\text{max}}$ denotes the total match score from the optimal matching.

# E. ToolRL Hyperparameters Setting

*Table 6.* GDPO verl training configuration. All hyperparameter settings are kept identical to those used in ToolRL(Qian et al., 2025).

| Parameter | Value |
|---|---|
| trainer.total_epochs | 15 |
| data.train_batch_size | 512 |
| actor_rollout_ref.actor.ppo_mini_batch_size | 128 |
| data.max_prompt_length | 2048 |
| actor_rollout_ref.actor.optim.lr | 1.00E-06 |
| actor_rollout_ref.rollout.n | 4 |
| algorithm.kl_ctrl.kl_coef | 0.001 |

# F. Math/Coding Reasoning Hyperparameters Setting

*Table 7.* GDPO verl training configuration

| Parameter | Value |
|---|---|
| data.train_batch_size | 512 |
| actor_rollout_ref.actor.ppo_mini_batch_size | 64 |
| actor_rollout_ref.actor.ppo_epochs | 1 |
| data.max_prompt_length | 1024 |
| actor_rollout_ref.actor.optim.lr | 1.00E-06 |
| actor_rollout_ref.rollout.temperature | 1 |
| actor_rollout_ref.rollout.n | 16 |
| actor_rollout_ref.actor.clip_ratio_low | 0.2 |
| actor_rollout_ref.actor.clip_ratio_high | 0.28 |
| algorithm.filter_groups.enable | TRUE |
| algorithm.filter_groups.metric | seq_reward |
| actor_rollout_ref.actor.kl_loss_coef | 0.0005 |
| actor_rollout_ref.actor.kl_loss_type | mse |

## G. Training curves of GRPO and GDPO when training DeepSeek-R1-1.5B/7B and Qwen3-4B-Instruct with $\mathcal{R}_{\text{length}}$ and $\mathcal{R}_{\text{correct}}$ on math reasoning data.

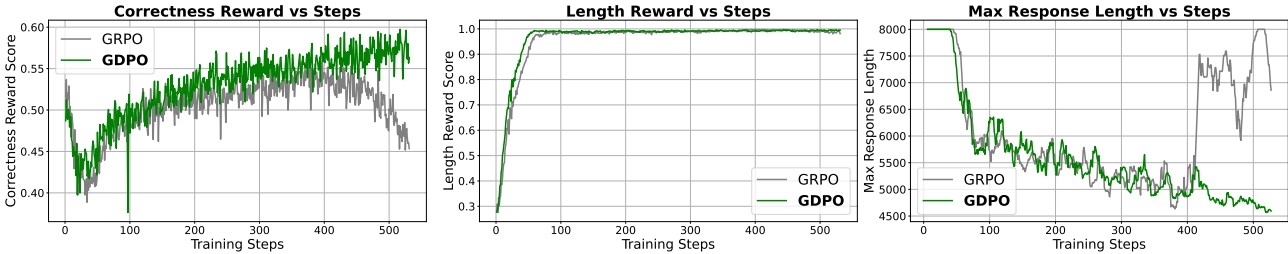

*Figure 8.* Training behavior of GRPO and GDPO on DeepSeek-R1-1.5B across correctness reward, length reward, and maximum batch response length. Both methods rapidly maximize the length reward, briefly suppressing correctness, yet GDPO subsequently recovers it and surpasses GRPO. After roughly 400 steps, GRPO's correctness score declines and its length-constraint violations increase, as reflected by rising maximum response lengths. In contrast, GDPO continues to improve correctness while steadily improving the control over response length.

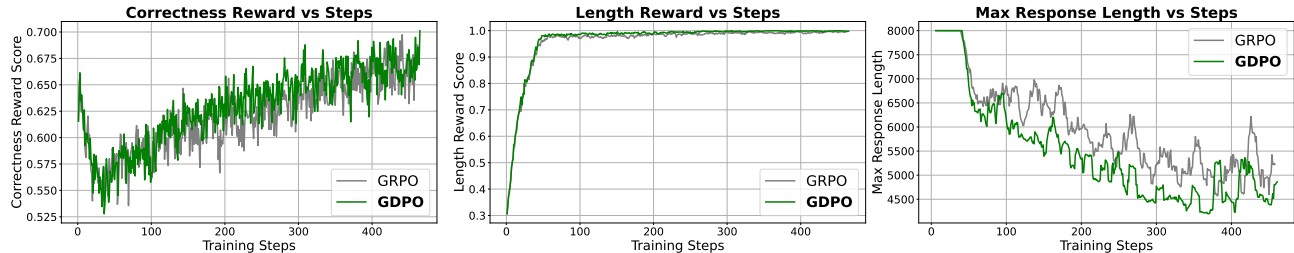

*Figure 9.* Training behavior of GRPO and GDPO when optimizing DeepSeek-R1-7B across correctness reward, length reward, and maximum batch response length on math reasoning data. We can see that GDPO maintains improving correctness and better adherence to length constraints over GRPO.

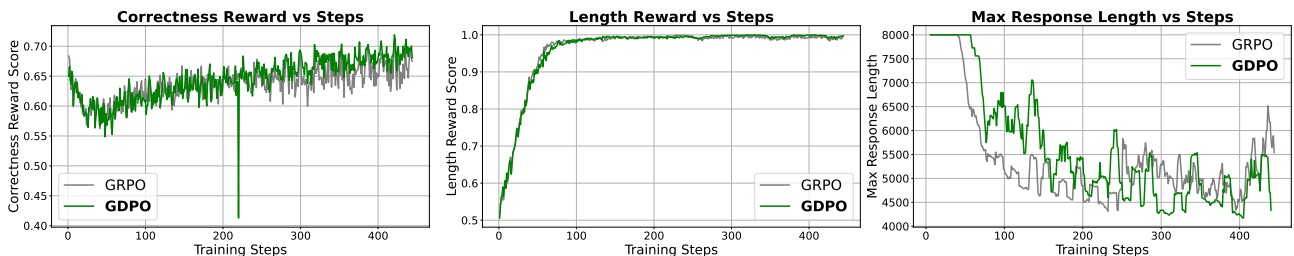

*Figure 10.* Training behavior of GRPO and GDPO when optimizing Qwen3-4B-Instruct across correctness reward, length reward, and maximum batch response length on math reasoning data. We can see that GDPO maintains improving correctness and better adherence to length constraints over GRPO.

## H. Comparison of GRPO and GDPO-trained DeepSeek-R1-1.5B/7B and Qwen3-4B-Instruct models across mathematical reasoning benchmarks.

*Table 8.* Comparison of GRPO and GDPO-trained DeepSeek-R1-1.5B/7B and Qwen3-4B-Instruct models on Pass@1 accuracy and the proportion of responses exceeding the length constraint across mathematical reasoning benchmarks.

| | | DeepSeek-R1-1.5B | | | DeepSeek-R1-7B | | | Qwen3-4B-Instruct | | |
| --- | --- | --- | --- | --- | --- | --- | --- | --- | --- | --- |
| | | - | GRPO | GDPO | - | GRPO | GDPO | - | GRPO | GDPO |
| MATH | Acc ↑ | 84.3% | 83.6% | **86.2%** | 93.6% | **94.1%** | 93.9% | 94.6% | 93.9% | 93.9% |
| | Exceed ↓ | 35.0% | 1.5% | **0.8%** | 26.0% | 0.5% | **0.1%** | 11.3% | 0.8% | **0.1%** |
| AIME | Acc ↑ | 29.8% | 23.1% | **29.4%** | 55.4% | 50.2% | **53.1%** | 63.7% | 54.6% | **56.9%** |
| | Exceed ↓ | 91.5% | 10.8% | **6.5%** | 85.6% | 2.1% | **0.2%** | 71.3% | **2.5%** | **0.1%** |
| AMC | Acc ↑ | 62.0% | 64.5% | **69.0%** | 82.9% | 83.8% | **84.0%** | 84.5% | **85.2%** | 84.3% |
| | Exceed ↓ | 67.5% | 3.2% | **2.3%** | 57.2% | 0.6% | **0.3%** | 33.9% | 0.7% | **0.1%** |
| Minerva | Acc ↑ | 38.41.% | 43.5% | **44.0%** | 49.8% | 53.2% | **53.8%** | 50.7% | **52.4%** | 51.9% |
| | Exceed ↓ | 51.4% | 1.7% | **0.3%** | 41.8% | 0.2% | **0.1%** | 9.1% | 0.3% | **0.1%** |
| Olympiad | Acc ↑ | 44.1% | 44.3% | **46.6%** | 58.2% | **60.2%** | 59.7% | 65.7% | 66.8% | **67.5%** |
| | Exceed ↓ | 70.1% | 2.6% | **1.9%** | 60.6% | 1.1% | **0.4%** | 41.3% | 1.6% | **1.0%** |

# I. Average accuracy and exceed-length ratios under varying length reward weights

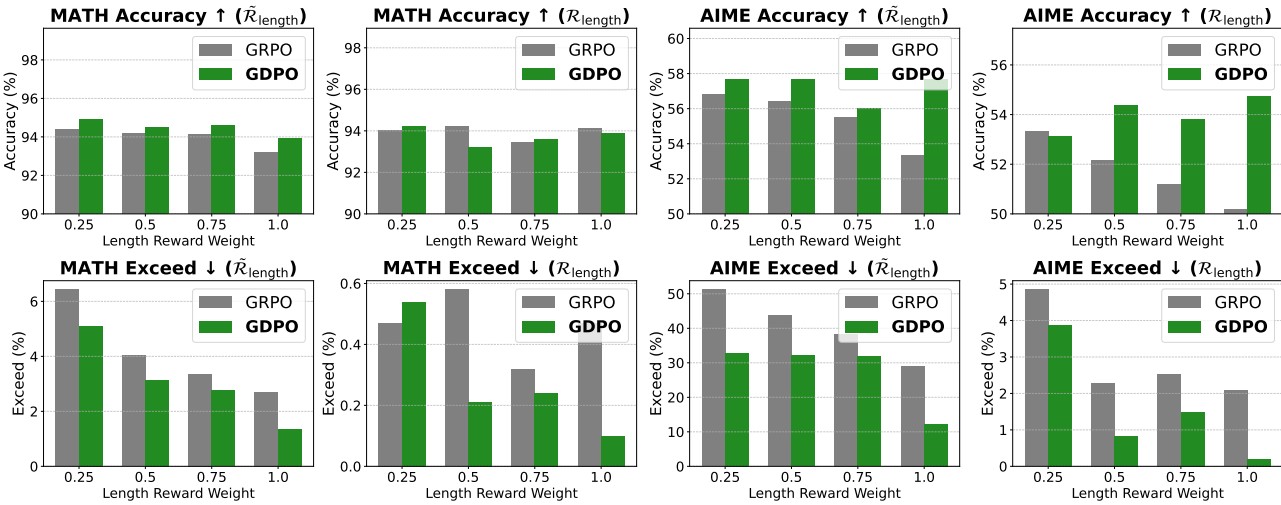

*Figure 11.* Average accuracy and exceed-length ratios for GRPO/GDPO-trained DeepSeek-R1-7B models under varying length reward weights $\{1.0, 0.75, 0.5, 0.25\}$, with and without the conditioned length reward $\tilde{\mathcal{R}}_{\text{length}}$, on mathematical reasoning tasks.

## J. Comparison of GRPO and GDPO trained with and without the conditioned length reward

*Table 9.* Comparison of GRPO and GDPO DeepSeek-R1-7B models, with and without the conditioned length reward $\tilde{\mathcal{R}}_{\text{length}}$, on Pass@1 accuracy and the ratio of outputs exceeding the length constraint across mathematical reasoning benchmarks.

| | | DeepSeek-R1-7B | | | | |
| --- | --- | --- | --- | --- | --- | --- |
| | | - | $\mathcal{R}_{\text{length}}$ | | $\tilde{\mathcal{R}}_{\text{length}}$ | |
| | | - | GRPO | GDPO | GRPO | GDPO |
| MATH | Acc ↑ | 93.6% | **94.1%** | 93.9% | 93.2% | **93.9%** |
| | Exceed ↓ | 26.0% | 0.5% | **0.1%** | 2.7% | **1.4%** |
| AIME | Acc ↑ | 55.4% | 50.2% | **53.1%** | 53.3% | **57.7%** |
| | Exceed ↓ | 85.6% | 2.1% | **0.2%** | 29.2% | **12.3%** |
| AMC | Acc ↑ | 82.9% | 83.8% | **84.0%** | 82.9% | **85.9%** |
| | Exceed ↓ | 57.2% | 0.6% | **0.3%** | 8.6% | **3.8%** |
| Minerva | Acc ↑ | 49.8% | 53.2% | **53.8%** | 53.2% | **53.4%** |
| | Exceed ↓ | 41.8% | 0.2% | **0.1%** | 2.5% | **1.0%** |
| Olympiad | Acc ↑ | 58.2% | **60.2%** | 59.7% | 59.1% | **60.8%** |
| | Exceed ↓ | 60.6% | 1.1% | **0.4%** | 10.3% | **6.2%** |

From Table. 9, we observe that using $\tilde{\mathcal{R}}_{\text{length}}$ leads to a larger increase in the average length-exceeding ratio for both GRPO and GDPO compared with merely adjusting the weight $w_{\text{length}}$ of $\mathcal{R}_{\text{length}}$, indicating a more effective relaxation of the length constraint. However, GRPO fails to convert this relaxed constraint into meaningful accuracy improvements. In contrast, GDPO prioritizes the correctness reward more effectively and achieves more consistent accuracy improvement over training without $\tilde{\mathcal{R}}_{\text{length}}$, while introducing substantially smaller increases in length violations. For instance, using $\tilde{\mathcal{R}}$length with GDPO yields a 4.4% accuracy improvement on AIME with a 16.9% reduction in length-exceeding ratio, and a 3% accuracy gain on AMC with a 4.8% reduction in length violations compared with using GRPO with the same reward.

## K. Comparison of GRPO/GDPO under varying length reward weights with and without the conditioned length reward

*Table 10.* Comparison of GRPO/GDPO trained DeepSeek-R1-7B models under varying length reward weights $\{1.0, 0.75, 0.5, 0.25\}$ with the normal length reward $\mathcal{R}_{\text{length}}$ on math reasoning tasks

| | Length Reward Weight | MATH ↑ | Exceed ↓ | AIME ↑ | Exceed ↓ | Amc ↑ | Exceed ↓ | Minerva ↑ | Exceed ↓ | Olympiad ↑ | Exceed ↓ | Avg Acc ↑ | Avg Exceed ↓ |
|---|---|---|---|---|---|---|---|---|---|---|---|---|---|
| DeepSeek-R1-7B | - | 93.6% | 26.0% | 55.4% | 85.6% | 82.9% | 57.2% | 49.8% | 41.8% | 58.2% | 60.6% | 68.0% | 54.2% |
| GRPO-$\mathcal{R}_{\text{length}}$ | 0.25 | 94.0% | 0.5% | 53.3% | 4.9% | 83.8% | 0.6% | 53.2% | 1.9% | 59.8% | 2.4% | 68.8% | 2.1% |
| | 0.5 | 94.2% | 0.6% | 52.1% | 2.3% | 83.2% | 0.8% | 53.9% | 0.2% | 60.2% | 0.8% | 68.7% | 0.9% |
| | 0.75 | 93.5% | 0.3% | 51.2% | 2.5% | 83.0% | 0.5% | 53.4% | 0.1% | 58.2% | 1.5% | 67.8% | 1.0% |
| | 1.0 | 94.1% | 0.5% | 50.2% | 2.1% | 83.8% | 0.6% | 53.2% | 0.2% | 60.2% | 1.1% | 68.3% | 0.9% |
| GDPO-$\mathcal{R}_{\text{length}}$ | 0.25 | 94.2% | 0.5% | 54.7% | 3.9% | 84.5% | 1.4% | 54.1% | 1.1% | 59.1% | 1.3% | 69.3% | 1.6% |
| | 0.5 | 93.2% | 0.2% | 53.8% | 0.8% | 84.1% | 0.4% | 53.3% | 0.2% | 58.5% | 1.1% | 68.6% | 0.5% |
| | 0.75 | 93.6% | 0.2% | 54.4% | 1.5% | 83.4% | 0.2% | 53.4% | 0.3% | 58.8% | 0.6% | 68.7% | 0.5% |
| | 1.0 | 93.9% | 0.1% | 53.1% | 0.2% | 84.0% | 0.3% | 53.8% | 0.1% | 59.3% | 0.4% | 68.8% | 0.2% |

*Table 11.* Comparison of GRPO/GDPO trained DeepSeek-R1-7B models under varying length reward weights $\{1.0, 0.75, 0.5, 0.25\}$ with the conditioned length reward $\tilde{\mathcal{R}}_{\text{length}}$ on math reasoning tasks

| | Length Reward Weight | MATH ↑ | Exceed ↓ | AIME ↑ | Exceed ↓ | Amc ↑ | Exceed ↓ | Minerva ↑ | Exceed ↓ | Olympiad ↑ | Exceed ↓ | Avg Acc ↑ | Avg Exceed ↓ |
|---|---|---|---|---|---|---|---|---|---|---|---|---|---|
| DeepSeek-R1-7B | - | 93.6% | 26.0% | 55.4% | 85.6% | 82.9% | 57.2% | 49.8% | 41.8% | 58.2% | 60.6% | 68.0% | 54.2% |
| GRPO-$\tilde{\mathcal{R}}_{\text{length}}$ | 0.25 | 94.4% | 6.4% | 56.8% | 51.2% | 85.8% | 18.2% | 54.5% | 6.5% | 61.2% | 18.8% | 70.6% | 20.2% |
| | 0.5 | 94.2% | 4.0% | 56.5% | 43.8% | 85.9% | 13.6% | 54.8% | 7.1% | 60.6% | 17.2% | 70.4% | 17.1% |
| | 0.75 | 94.1% | 3.4% | 55.5% | 38.3% | 85.2% | 11.2% | 54.3% | 5.7% | 60.8% | 14.3% | 70.0% | 14.6% |
| | 1.0 | 93.2% | 2.7% | 53.3% | 29.2% | 82.9% | 8.6% | 53.2% | 2.5% | 59.1% | 10.3% | 68.3% | 10.7% |
| GDPO-$\tilde{\mathcal{R}}_{\text{length}}$ | 0.25 | 94.9% | 5.1% | 57.7% | 32.8% | 86.2% | 11.2% | 54.9% | 5.1% | 62.1% | 14.6% | 71.2% | 13.8% |
| | 0.5 | 94.5% | 3.1% | 57.7% | 32.1% | 85.8% | 9.3% | 54.1% | 3.9% | 61.6% | 13.3% | 70.8% | 12.3% |
| | 0.75 | 94.6% | 2.8% | 56.0% | 31.9% | 86.9% | 10.3% | 53.0% | 2.6% | 61.3% | 12.8% | 70.4% | 12.1% |
| | 1.0 | 93.9% | 1.4% | 57.7% | 12.3% | 85.9% | 3.8% | 53.4% | 1.0% | 60.8% | 6.2% | 70.4% | 4.9% |

