# OpenReview forum: "GDPO: Group reward-Decoupled Normalization Policy Optimization for Multi-reward RL Optimization"
_ICML.cc/2026/Conference — ICML 2026 regular_

### Official Review · Reviewer_kJVt · 2026-02-19

**Soundness:** 3
**Presentation:** 3
**Significance:** 2
**Originality:** 2
**Overall Recommendation:** 3
**Confidence:** 4

**Summary:**

The paper studies multi-reward RL optimization when using GRPO with summed rewards. It argues that group-wise normalization on aggregated rewards can map many distinct reward combinations to the same (or very few) advantage values, reducing the resolution of the learning signal and sometimes causing unstable training. To address this, the authors propose GDPO: compute group-wise normalized advantages separately for each reward, sum them, and then apply an additional batch-wise advantage normalization to keep the advantage scale stable as the number of rewards increases. Experiments on tool calling (format + correctness), math reasoning (accuracy + length constraint), and coding (pass rate + length + bug-free) show GDPO generally improves reward curves and downstream metrics over GRPO.

**Compliance With Llm Reviewing Policy:**

Affirmed.

**Final Justification:**

This paper studies a practically relevant issue in multi-reward LLM post-training: when GRPO is applied to the sum of multiple rewards, the subsequent group-wise normalization can compress distinct reward combinations into identical or overly similar advantages, reducing the resolution of the learning signal. The proposed GDPO method addresses this by normalizing each reward separately before aggregation and then applying an additional batch-wise normalization to stabilize the overall advantage scale. The paper is generally clear, the motivation is easy to follow, and the problem itself is relevant to current multi-reward RLHF/RLVR practice.

The rebuttal addressed most of my original technical concerns. In particular, the added “GRPO + additional BN” control baseline and the extra ablations substantially clarify that the observed gains are not simply due to Eq. (5). The newly provided multi-seed results for both math and coding strengthen the robustness of the empirical claims. The authors also clarified previously underspecified implementation details, including zero-variance handling, the interpretation of reward weighting under GDPO, and the overlap-checking procedure for the training/evaluation data. I appreciate the authors’ detailed and constructive responses, which improved my confidence in the soundness and reproducibility of the work.

That said, my final score remains unchanged mainly because I still view the paper’s methodological novelty as limited. I agree with Reviewer z4pa’s concern that the central notion of group reward-decoupled normalization is close in spirit to prior work such as BNPO, where related advantage decomposition ideas already appear, and that the main methodological distinction here is the additional batch-wise normalization term. In this sense, I see the paper’s strongest contribution as a useful diagnosis and empirical clarification of a multi-reward GRPO issue, together with a practical refinement, rather than a substantially new optimization framework. I also think the final paper should position itself more explicitly relative to BNPO and other normalization-based variants. The added clarification around Fig. 2(b) is helpful, but it does not materially change my assessment of originality/significance.

Overall, after taking both the paper and the rebuttal into account, I view the submission as technically reasonable, clearly motivated, and improved by the authors’ responses. However, when weighing soundness, clarity, originality, and significance together, I remain at 3 because the contribution still feels more incremental than I would typically favor for acceptance in its current form. I thank the authors for the careful rebuttal and for addressing my questions in detail.

**Key Questions For Authors:**

1) Control baseline: What happens if you add the same batch-wise advantage whitening (Eq. 5 / masked_whiten) to GRPO (on the summed-reward advantage)? If GDPO still wins, the decoupling is the main driver; if not, the paper’s contribution should be reframed.
2) Robustness: For math and coding (Tables 2–3, 7–10), are results from a single run or multiple seeds? Please report at least 3 seeds (mean ± std or CI). If gains disappear under re-runs, the conclusion about stability should be weakened.
3) Edge cases: How do you handle Eq. 3 when a reward has zero within-group variance (all rollouts get the same reward value)? Different choices (ε in denominator vs returning zero advantage) can materially affect training stability.
4) Objective interpretation: Since per-reward standardization changes relative scaling across rewards per question, what objective is GDPO effectively optimizing compared to the original weighted sum of rewards? If the method implicitly reweights rewards, how should users choose weights to match desired trade-offs?
5) Data contamination: Did you check overlaps between training datasets and evaluation benchmarks (e.g., DeepScaleR-Preview vs MATH/AIME/AMC; Eurus-2-RL vs PRIME validation)? If overlaps exist, reported generalization may be overestimated.

**Limitations:**

The paper does not sufficiently discuss limitations. Suggested additions:
1) Clarify that GDPO changes effective reward scaling per question and may deviate from optimizing the raw weighted sum objective.
2) Discuss computational overhead as the number of rewards grows (extra advantage computations per reward).
3) Report stability variance across seeds, and discuss failure modes (e.g., sparse/constant rewards leading to zero variance).
4) State assumptions and limitations regarding correlated rewards and scaling beyond 3 reward signals.

**Strengths And Weaknesses:**

Strengths:
1) Clear motivation and illustration of the potential “advantage resolution” issue with GRPO under multiple rewards (Section 2, Figure 2).
2) The method is simple and easy to integrate; the paper provides explicit equations (Eq. 3–5) and an implementation sketch (Appendix C).
3) Experiments cover multiple tasks and model scales (Sections 4.1–4.3; Tables 1–3, 7–8), suggesting reasonable generality within LLM-RL settings.
4) Tool-calling experiments include 5-run statistics (median + IQR curves and averaged results), which is stronger than many RLHF-style submissions (Figure 3, Table 1).
5) Additional analysis on reward priority and “difficulty imbalance” via conditioned length reward is useful (Section 4.2.1; Figures 5–6; Tables 8–10).

Weaknesses:
1) Attribution is unclear because GDPO introduces an extra batch-wise advantage whitening step (Eq. 5; Appendix C), while the GRPO baseline is not clearly controlled with the same whitening. A key baseline “GRPO + the same batch-wise whitening” is missing.
2) Novelty appears incremental: the core change is per-reward normalization plus whitening, and the relationship to existing GRPO variants that already use normalization tricks is not fully clarified (Related Work vs Method).
3) Math and coding results do not report multi-seed variance/CI, so stability and significance are hard to assess beyond a single run (Tables 2–3, 7–10; Figures 4, 8–10).
4) Important implementation edge cases are not specified in the main text (e.g., how Eq. 3 handles zero within-group variance for a reward; exact masking/token-vs-sequence statistics for Eq. 5).
5) Potential dataset overlap/contamination checks between training and evaluation benchmarks are not discussed (Sections 4.1–4.3).

---

> ### Author Rebuttal · Authors · 2026-03-30
>
> **W1/Q1**
> | Qwen2.5-1.5B |Live Acc ↑|Multi Turn Acc↑|Non_Live Acc↑|Avg Acc↑|Format ↑ |
> |-|-|-|-|-|-|
> | GRPO | 50.63% | 2.04%  | 37.87%  | 30.18% | 76.33% |
> | GDPO | 55.36% | 2.50%  | 40.58%  | 32.81%  | 80.66% |
> | GRPO w/ additional BN | 49.97%  | 2.12% | 37.45%  | 29.85% | 75.86%|
>
> We agree this is an important control, and we therefore add GRPO + the same batch-wise normalization used in GDPO. As shown above, this variant performs almost identically to standard GRPO (30.18% → 29.85% Avg Acc; 76.33% → 75.86% Format) and remains clearly below GDPO (32.81%, 80.66%).
>
> This shows that GDPO’s gains are not simply due to the extra batch-wise normalization. This is also consistent with our distinct-group-count analysis: batch-wise normalization only rescales advantages and does not increase the number of distinct groups. The key improvement comes from per-reward normalization before aggregation.
>
> **W2**
>
> Our contribution is not merely a normalization tweak, but the identification of a reward-collapse issue in multi-reward GRPO, where distinct reward combinations can map to identical advantages after aggregation and normalization. We will revise the paper to better position GDPO relative to other related works and highlight that our contribution includes both (i) diagnosing the reward-collapse issue and (ii) proposing a principled solution.
>
> **W3/Q2/L3**
>
> For the math and coding experiments, we use three seeds and report results from checkpoints selected by a fixed evaluation criterion, rather than seed averages.
>
> For math, we select the checkpoint with the best AIME performance, since AIME is the most challenging task to improve reasoning efficiency without accuracy degradation. The same protocol is applied to both GDPO and all baselines.
>
> | Method | MATH Acc ↑ | MATH Exceed ↓ | AIME Acc ↑ | AIME Exceed ↓ |
> |-|-|-|-|-|
> | 1.5B-GRPO | 83.4% ± 0.012  | 1.89% ± 0.018 | 21.98% ± 0.007 | 12.11% ± 0.017 |
> | 1.5B-GDPO | 85.9% ± 0.009 | 0.92%  ± 0.011| 28.84% ± 0.010 | 7.3% ± 0.009|
> | 7B-GRPO | 93.4% ± 0.006 | 0.71% ± 0.008 | 48.9% ± 0.008 | 3.3% ± 0.018 |
> | 7B-GDPO | 93.8% ± 0.002 | 0.15%  ± 0.003 | 52.7% ± 0.009 | 0.6% ± 0.012 |
>
> To provide robustness evidence, we additionally report 3-seed averages for the 1.5B/7B math models (table above), where the overall trend remains consistent, and GDPO continues to outperform GRPO.
>
> For coding, we select the checkpoint with the best average pass rate across tasks, again using the same protocol for both methods. We will clarify this reporting protocol and include multi-seed mean ± std in the appendix.
>
> **W4/Q3**
>
> (1) Zero within-group variance in Eq. (3)
>
> This is not unique to GDPO and can also occur in GRPO when all rollouts in a group receive identical rewards. In our experiments (Sec 4.2 and 4.3), we follow the DAPO[1] training setup and use dynamic sampling to filter out training groups with zero within-group variance. Importantly, with dynamic sampling, GDPO still consistently outperforms GRPO, so the improvements are not due to special handling of this edge case.
>
> [1] DAPO: An Open-Source LLM Reinforcement Learning System at Scale
>
> (2) Masking / token-vs-sequence statistics for Eq. (5)
>
> Eq. (5) itself operates on the final rollout-level advantages after reward aggregation, and therefore does not depend on token masking or on whether the downstream policy objective uses token-level [1] or sequence-level averaging. In other words, Eq. (5) is simply a batch-wise normalization over the computed advantages.
>
> **Q5/L1**
>
> GDPO makes reward weighting more direct and interpretable: the final advantage is a weighted sum of the independently normalized reward advantages, so the weights directly control each reward’s contribution to the training signal.
>
> In contrast, in GRPO the weights are applied before normalization, and group normalization can distort their intended effect. Consistent with this, Tables 9–10 show that GDPO achieves a better accuracy–efficiency trade-off across weight settings.
>
> **W5/Q6**
>
> For both math and coding, we follow existing public training/evaluation setups from prior work and do not introduce benchmark-specific data collection that would create new contamination concerns.
>
> Specifically, the math setup follows [2,3,4], and the coding setup follows [5]. All datasets and benchmarks are public / open-sourced, and to our knowledge, there are no known contamination issues associated with these standard setups.
>
> [2] Learn to Reason Efficiently with Adaptive Length-based Reward Shaping (ICLR 2026)
>
> [3] AdaptThink: Reasoning Models Can Learn When to Think (ACL 25)
>
> [4] DLER: Doing Length pEnalty Right
>
> [5] ProRL: Prolonged Reinforcement Learning Expands Reasoning Boundaries in Large Language Models (NIPS 25)
>
> **L2**
>
> The additional overhead of GDPO is negligible, since the main bottleneck in RL training is rollout generation, not advantage computation. For example, on the tool-calling task, wall-clock time per training step is 12.5s for GDPO vs. 12.3s for GRPO.

---

> > ### Author Rebuttal · Reviewer_kJVt · 2026-04-02
> >
> > I have read the rebuttal carefully. Several of my original concerns are meaningfully addressed. In particular, the added control baseline (“GRPO + the same batch-wise normalization”) makes it much more convincing that the gains are not simply due to Eq. (5), and the additional ablations (“GDPO w/o Batch Norm” and “GDPO w/ Group Norm”) further strengthen the attribution to reward decoupling rather than a generic whitening effect. The additional math results across 3 seeds, as well as the rollout ablation, also strengthen the empirical case.
> >
> > However, I still have a few follow-up questions before considering my concerns fully resolved:
> >
> > 1. For coding, the rebuttal says 3-seed mean ± std results will be added in the appendix, but these numbers are not currently shown. Since Table 3 is the main evidence for the 3-reward setting, could you please provide the multi-seed coding results explicitly?
> >
> > 2. For Eq. (3), the zero-variance explanation is still a bit unclear to me. The response mentions dynamic sampling, but please clarify whether zero-variance filtering is applied per reward component or only on the aggregate reward, and what the exact fallback is when one reward component has zero within-group variance but other components do not (e.g., epsilon in the denominator vs. setting that component’s normalized advantage to zero vs. filtering the group).
> >
> > 3. The clarification on reward weighting is helpful, but I still think the paper should state more explicitly that per-reward standardization induces adaptive per-question rescaling relative to the raw weighted-sum objective. A short explanation of how users should interpret/tune weights under this behavior would help.
> >
> > 4. On contamination/overlap, the rebuttal says the work follows standard public setups and that there are no known issues, but it is still unclear whether any explicit overlap audit was performed. Please clarify whether an actual check was done, or whether this is based on following prior public protocols.
> >
> > Also, clearer positioning relative to BNPO / other normalization-based variants would help me better assess the originality claim.
> >
> > Overall, the rebuttal improves my assessment, especially on attribution and the math-side robustness, but the above points remain partially unresolved for me.

---

> > > ### Author Response · Authors · 2026-04-03
> > >
> > > **(1) Multi-seed coding results**
> > >
> > > We agree that Table 3 is central to our claim in the 3-reward setting, and we therefore provide the requested 3-seed mean ± std results below for DeepSeek-R1-7B on coding reasoning.
> > > |Benchmark|Metric|$R_{Pass}$ + $R̃_{length}$ GRPO₂-obj | $R_{Pass}$ + $R̃_{length}$ GDPO₂-obj |$R_{Pass}$ + $R̃_{length}$ + $R_{Bug}$ GRPO₃-obj|$R_{Pass}$ + $R̃_{length}$ + $R_{Bug}$ GDPO₃-obj|
> > > |-|-|-|-|-|-|
> > > |Apps|Pass↑|66.5% ± 0.007|**67.9% ± 0.003**|67.1% ± 0.008|**67.6% ± 0.001**|
> > > ||Exceed↓|5.8% ± 0.007|**5.2% ± 0.001**| 11.8% ± 0.005 |**8.8% ± 0.003**|
> > > ||Bug↓|25.8% ± 0.008|**24.1% ± 0.007**| 21.4% ± 0.013 |**19.7% ± 0.011**|
> > > |Codecontests|Pass ↑|62.6% ± 0.005| **65.1% ± 0.008** |64.6% ± 0.007|**65.3% ± 0.002**|
> > > ||Exceed↓|14.8% ± 0.006|**14.6% ± 0.003**|20.1% ± 0.007|**16.6% ± 0.008**|
> > > ||Bug↓|15.4% ± 0.012|**14.4%±0.011**|4.63% ± 0.007|**3.5% ± 0.009**|
> > >
> > > These results are consistent with the main paper’s single-checkpoint comparisons and show that GDPO’s advantage is robust across random seeds. Due to space constraints, we will include the multi-seed coding results (contest and TACO) in the final paper.
> > >
> > > **(2) Zero-variance handling**
> > >
> > > In our setup, zero-variance filtering is applied only to the aggregated reward for both GRPO and GDPO. Concretely, dynamic sampling checks whether the group-level aggregated reward has non-zero variance; if it is constant across all rollouts, the question is skipped during training.
> > >
> > > For GDPO, we do not apply additional filtering at the level of individual reward components when one of them has zero within-group variance, since in that case the appropriate behavior is simply for that component to contribute nothing to the update. Specifically, If one reward component has zero within-group variance, its normalized advantage becomes zero while other reward components that do vary still contribute valid non-zero advantages. Regarding implementation, we always include epsilon in the denominator, following standard GRPO implementations such as VERL [1] and TRL [2], regardless of whether dynamic sampling is applied.
> > >
> > > **(3) GDPO reward weighting**
> > >
> > > We agree this point should be stated more explicitly. In GDPO, each reward is normalized independently per question, and as a result, the weights determine the relative importance of each objective after normalization, rather than being tied to the absolute scale of the raw rewards, as in GRPO. In practice, users should interpret and tune the weights as direct trade-off parameters between objectives, and rescaling the raw rewards before normalization does not affect their contribution in the same way as in GRPO.
> > >
> > > **(4) Data overlap**
> > >
> > > Our response is not based solely on following prior works; we also conducted an explicit overlap check. For math reasoning experiment, the training data includes sources such as AIME 1984–2023 and AMC prior to 2023., while evaluation is performed on held-out benchmarks including AIME 2024, AMC 2023, ensuring no overlap. For the coding task, we follow the public split from PRIME [3], which similarly avoids contamination.
> > >
> > > **(5) Clearer positioning relative to other normalization-based variants**
> > >
> > > We agree that the positioning relative to BNPO and other normalization-based variants should be clearer, and we will revise the Related Work section accordingly.
> > >
> > > BNPO is the closest in form, since it also decomposes advantages across rewards before combining them. However, BNPO focuses on **single binary reward Beta-based normalization**, and does not analyze or resolve the multi-reward-collapse issue we identify here. Our contribution is therefore not only per-reward normalization, but also the identification of this issue and a principled two-stage design that combines per-reward group-wise normalization with post-aggregation batch-wise normalization, where the two components serve different purposes.
> > >
> > > Related batch-normalization-based variants such as [4] and Reinforce++ [5] also differ in focus, as they primarily study single-reward RL and use batch-normalization to reduce variance and improve training stability. In contrast, GDPO applies batch-wise advantages normalization to ensure that the numerical scale of the final advantage sum remains stable and does not grow as additional rewards are introduced. We will make this distinction more explicit in the revised paper.
> > >
> > > **4/8 Response to Reviewer's final justification**
> > >
> > > One final clarification: while BNPO is closest in form, it does not analyze the multi-reward collapse issue we study and, to the best of our understanding, mainly focuses on binary-reward settings. In contrast, our work studies broader continuous / diverse multi-reward settings, with both theoretical and empirical analysis.
> > >
> > > [1] verl-project/verl
> > >
> > > [2] huggingface/trl
> > >
> > > [3] PRIME-RL/Eurus-2-RL-Data
> > >
> > > [4] What matters in on-policy reinforcement learning? a large-scale empirical study
> > >
> > > [5] REINFORCE++: Stabilizing Critic-Free Policy Optimization with Global Normalization

---

### Official Review · Reviewer_z4pa · 2026-03-02

**Soundness:** 2
**Presentation:** 3
**Significance:** 2
**Originality:** 2
**Overall Recommendation:** 2
**Confidence:** 4

**Summary:**

This paper analyzes why applying Group Relative Policy Optimization (GRPO) directly to summed multi-reward signals can lead to overly compressed advantage estimates and unstable training. It shows that group-wise normalization on aggregated rewards erases meaningful distinctions between different reward combinations. To remedy this, the authors introduce Group reward-Decoupled normalization Policy Optimization (GDPO), which performs normalization separately for each reward before aggregation. Experiments on tool use, mathematical reasoning, and coding demonstrate that GDPO yields more stable optimization and better accuracy–constraint trade-offs than GRPO.

**Compliance With Llm Reviewing Policy:**

Affirmed.

**Final Justification:**

Given the strong similarity of the 'advantage decomposition' in BNPO, I will maintain my score.

**Key Questions For Authors:**

1. Which metric is used to select the best checkpoint for the experiments reported in Table 1?

2. I do not observe a clear difference between GRPO w/o std and GDPO. If the standard deviation term in Eq. (3) is removed, GDPO becomes equivalent to GRPO w/o std. Does GDPO always retain the standard deviation term in Eq. (3)?

3. To improve the stability of the final advantage, would it be appropriate to replace the sum aggregation in Eq. (4) with a mean aggregation?

**Limitations:**

yes

**Strengths And Weaknesses:**

Strengths:

1. The paper provides a simple yet convincing theoretical and illustrative analysis showing how GRPO collapses distinct multi-reward configurations into identical advantages, which is both intuitive and practically relevant.

2. The method is evaluated across diverse tasks (tool use, math reasoning, coding), multiple model families, and both two- and three-reward settings, consistently demonstrating improved stability and better multi-objective trade-offs.

----
Weaknesses:

1. The concept of group reward–decoupled normalization has been proposed in BNPO [1], where it is referred to as Advantage Decomposition. The primary distinction between GDPO and BNPO lies in the inclusion of an additional batch-wise normalization term (Eq. (5)). Consequently, the contribution of GDPO in terms of methodological novelty appears limited.

2. Figure 2(b) suggests that GDPO benefits more significantly from increases in both the number of rollouts and the number of rewards. However, this claim lacks empirical validation, as no experiments have been conducted to confirm these observations. Besides, could you provide the general formulas used for Figure 2(b) and clarify how the results are computed? This would help in understanding the specific strengths of GDPO.

[1] Xiao C, Zhang M, Cao Y. Bnpo: Beta normalization policy optimization[J]. arXiv preprint arXiv:2506.02864, 2025.

---

> ### Author Rebuttal · Authors · 2026-03-30
>
> **W1**
> ```
> The concept of group reward–decoupled normalization has been proposed in BNPO [1]....
> ```
> We thank the reviewer for pointing out BNPO. While there is some high-level similarity in treating rewards separately, we would like to emphasize that our contribution is not only the algorithmic design of GDPO, but also the identification of a reward-collapse issue in multi-reward GRPO, where distinct reward combinations can map to identical advantages after aggregation and normalization.
>
> GDPO is proposed as a targeted and principled fix: by normalizing each reward before aggregation, it preserves cross-reward distinctions that GRPO loses, while the final batch-wise normalization stabilizes scale without re-compressing these distinctions. Importantly, each component of GDPO is carefully motivated and empirically examined, rather than being an arbitrary normalization choice.
>
> We will revise the paper to better position GDPO relative to BNPO and highlight that our contribution includes both (i) diagnosing the reward-collapse issue and (ii) proposing a principled solution.
>
> **W2 (1)**
> ```
> Figure 2(b) suggests....... However, this claim lacks empirical validation, as no experiments have been conducted to confirm these observations.
> ```
> | Method |Rollout| MATH Acc ↑ | MATH Exceed ↓ | AIME Acc ↑ | AIME Exceed ↓ | AMC Acc ↑ | AMC Exceed ↓ |
> |--|-|-|-|-|-|-|-|
> | 1.5B   | -   | 84.3%    | 35.0%          | 29.8%      | 91.5%          | 62.0%     | 67.5%          |
> | GRPO  | 32 | 83.4%    | 1.4%          | 24.2%      | 11.8%          | 65.7%     | 4.2%          |
> | GDPO  | 32 | **87.5%**    | **0.5%**          | **31.5%**      | **7.1%**          | **70.9%**    | **2.1%**         |
> | GRPO  | 16 | 83.6%    | 1.5%          | 23.1%      | 10.8%          | 64.5%     | 3.2%          |
> | GDPO  | 16 | **86.2%**    | **0.8%**          | **29.4%**      | **6.5%**          | **69.0%**    | **2.3%**         |
> | GRPO  | 8 | 81.1%    | 2.4%          | 22.6%      | 11.4%          | 63.3%     | 3.8%          |
> | GDPO  | 8 | **85.4%**    | **1.3%**          | **29.3%**      | **7.2%**          | **67.4%**    | **3.1%**         |
> | GRPO  | 4 | 81.0%    | 1.9%          | 22.4%      | 12.8%          | 63.5%     | 4.1%          |
> | GDPO  | 4 | **84.7%**    | **1.2%**          | **28.2%**      | **6.8%**          | **67.5%**    | **2.6%**         |
>
> To provide empirical validation, we conduct a rollout ablation (4–32 rollouts) on math reasoning (table above). GDPO consistently outperforms GRPO across all settings, with gains generally increasing as the number of rollouts grows (e.g., AIME: +5.8 → +7.3, AMC: +4.0 → +5.2 from 4 to 32 rollouts). GDPO also consistently achieves a better accuracy–length trade-off.
>
> For the number of rewards, our coding experiments (2→3 rewards) already show consistent improvements. We agree that extending to 4+ rewards would further strengthen the paper, and we plan to include additional experiments in the revision using meaningful reward settings rather than arbitrary extra rewards.
>
> **W2 (2)**
> ```
> Besides, could you provide the general formulas used for Figure 2(b) ...?
> ```
>
> We provide pseudo-code for reproducing Fig. 2(b) here: https://anonymous.4open.science/r/gdpo_fig2-20BE/README.md. The same enumeration-and-counting procedure is used for both different numbers of rollouts/rewards simulations.
>
> **Q1**
>
> For Table 1, we evaluate the final saved checkpoint (no checkpoint selection) and report the average over 5 runs. We will clarify this in the revision.
>
> **Q2**
>
> The reviewer is correct that if the standard deviation term in Eq. (3) is removed, GDPO becomes equivalent to GRPO w/o std. With mean-centering only, “normalize each reward independently and then sum” is algebraically identical to “sum rewards first and then subtract the group mean,” due to the linearity of summation and mean.
>
> For this reason, the standard deviation term in Eq. (3) is essential to GDPO. It is precisely this per-reward variance normalization that prevents the method from degenerating to GRPO w/o std and allows GDPO to preserve more distinct advantage groups. This is also reflected in our distinct-group-count analysis, where GDPO yields substantially more distinct advantage groups than GRPO w/o std as the number of rollouts increases (see table below).
>
> | Distinct Advantage Groups | 2 Rollouts | 3 Rollouts | 4 Rollouts | 5 Rollouts | 6 Rollouts | 7 Rollouts | Rollout #8 |
> |-|-|-|-|-|-|-|-|
> | GRPO no STD | 3 | 6| 10 | 15 | 21 | 28 | 36   |
> | GDPO | 3   | 6 | 14| 26  | 38  | 59 | 81|
>
> Therefore, GDPO always retains the standard deviation term in all experiments. More broadly, this highlights that each component of GDPO is carefully motivated and necessary, rather than an arbitrary design choice.
>
> **Q3**
>
> Replacing the sum in Eq. (4) with a mean only introduces a constant rescaling and does not change the number of distinct advantage groups nor change the variance of advantage; mean aggregation provides no practical benefit.

---

> > ### Author Rebuttal · Reviewer_z4pa · 2026-04-04
> >
> > Thank you for your response. Given the strong similarity of the 'advantage decomposition' in BNPO, I will maintain my score.

---

> > > ### Author Response · Authors · 2026-04-04
> > >
> > > Thank you for the follow-up. We agree that BNPO is the closest related method in terms of high-level form, and we will revise the related work section of the paper to make this relationship clearer. At the same time, we would like to respectfully clarify why we believe the contribution remains original and meaningful in the ICML sense.
> > >
> > > Our paper’s main contribution is **not only the use of reward-wise decomposition**, but the identification and analysis of **a previously overlooked failure mode in multi-reward GRPO**, namely that **distinct reward combinations can collapse into identical group-relative advantages after aggregation and normalization**. To the best of our understanding, **this issue is not analyzed in BNPO**, nor is BNPO motivated by or evaluated around this failure mode.
> > >
> > > Moreover, our method is not simply “decompose and sum.” GDPO is a **carefully designed two-stage normalization scheme: per-reward group-wise normalization** is introduced to preserve cross-reward distinctions, while **post-aggregation batch-wise normalization** is introduced to stabilize the final advantage scale as the number of rewards grows. We also added controlled experiments showing that this second component is not incidental, but contributes a distinct and complementary effect to the final performance. In this sense, the contribution is not just a similar-looking decomposition step, but **a principled design tailored to the multi-reward GRPO setting**.
> > >
> > > More broadly, we believe the paper is original along the dimensions emphasized in the ICML reviewer guidelines, namely by providing a **new insight into an important limitation of GRPO**, deepening understanding of **multi-reward RL optimization**, and proposing a **targeted remedy supported by both theory and experiments**. We hope this clarifies why we believe the contribution is original and meaningful under the ICML criteria.

---

### Official Review · Reviewer_faQc · 2026-03-05

**Soundness:** 3
**Presentation:** 4
**Significance:** 3
**Originality:** 3
**Overall Recommendation:** 4
**Confidence:** 4

**Summary:**

Vanilla GRPO struggles to normalize distinct rollout reward combinations, causing them to collapse into identical advantage values. These identical advantages lead to information loss and poor learning learning. To solve this, the authors introduce Group reward-Decoupled Normalization Policy Optimization (GDPO), which normalizes individual rewards rather than the sum of rewards. This leads to many more configurations that yield a better learning signal. The authors show improvements over GRPO across 5 benchmarks, including AIME and agentic coding.

**Compliance With Llm Reviewing Policy:**

Affirmed.

**Final Justification:**

New experiments addressed most of my concerns.

**Key Questions For Authors:**

How do weighted rewards affect the performance of GDPO?

Addressing the concerns/weaknesses would lead me to increase the score.

**Limitations:**

Experiments are too few to really appreciate the findings.

**Strengths And Weaknesses:**

Strenghts
- The paper is well written and easy to follow.
- The problem in the paper is well motivated.
- The proposed solution is easy to implement within the existing pipelines.

Weakness
- The applicability of the approach is somewhat limited, as the problem highlighted occurs only when rewards are binary (or discrete), and continuous rewards won’t face the same issue.
- Results are marginally better, especially for larger models.
- Experiments are limited to only GRPO as a baseline. Why not try with other RLVR methods like CISPO or DAPO?

---

> ### Author Rebuttal · Authors · 2026-03-30
>
> **W1**
> ```
> The problem highlighted occurs only when rewards are binary, not continuous.
> ```
> We appreciate this observation. We agree that the exact combinatorial collapse shown in Sec. 2 is easiest to formalize for discrete rewards, where distinct reward tuples can be shown to map to the same normalized advantage. However, the broader issue addressed by GDPO is not limited to binary rewards.
> The key problem is that aggregating multiple rewards before normalization can wash out per-reward distinctions, regardless of whether the rewards are binary, discrete, or continuous. In the discrete case, this manifests as exact collapse (different reward combinations mapping to the same normalized advantage). In the continuous case, the same phenomenon appears more generally as a loss of reward-level resolution, even if the collisions are not always exact.
> We already evaluate GDPO in a continuous reward setting in Sec. 4.3, where the coding pass-rate reward $R_{\text{pass}} \in [0,1]$ is continuous. GDPO continues to outperform GRPO in this setting, which suggests that the applicability of the method extends beyond purely binary rewards. We will make this point clearer in the revised paper.
>
> **W2**
> ```
> Results are marginally better, especially for larger models.
> ```
> We agree that the relative gains are smaller for larger models. However, this trend is also observed in prior work using the same setup (e.g., [1],[2],[3]) and is not specific to GDPO.
>
> We further verify this with an additional single-reward GRPO (accuracy-only) experiment, where the 1.5B model again improves much more than the 7B model (e.g., AIME: +8.3% vs. +0.7%). This suggests the effect is mainly due to the training/evaluation setting rather than the method itself.
>
> Method | MATH Acc ↑ | MATH Exceed ↓ | AIME Acc ↑ | AIME Exceed ↓ | AMC Acc ↑ | AMC Exceed ↓ |
> |-|-|-|-|-|-|-|
> | 1.5B | 84.3% | 35.0% | 29.8% | 91.5% | 62.0% | 67.5% |
> | 1.5B (Only Accuracy reward) | 89.22% | 24.1% | 38.12% | 84.3% | 73.04% | 61.3% |
> | 7B | 93.6% | 26.0% | 55.4% | 85.6% | 82.9% | 57.2% |
> | 7B (Only Accuracy reward) | 94.3% | 29.7% | 56.1% | 74.2% | 86.1% | 52.3% |
>
>
> Importantly, GDPO still provides consistent improvements across model scales.
>
> [1] DLER: Doing Length pEnalty Right
>
> [2] Learn to Reason Efficiently with Adaptive Length-based Reward Shaping (ICLR 26)
>
> [3] AdaptThink: Reasoning Models Can Learn When to Think (ACL 25)
>
> **W3**
> ```
> Why not try with other RLVR methods like CISPO or DAPO?
> ```
> We would like to clarify that GDPO is orthogonal to the choice of policy optimization objective. GDPO only changes the advantage estimation, and can therefore be directly integrated into policy optimization methods such as GRPO, DAPO, or CISPO by replacing the advantage estimator. For example, given the DAPO policy optimization objective:
> $J_{\mathrm{DAPO}}(\theta)
> = \mathbb{E}\Big[
> \frac{1}{\sum_{i=1}^{G}|o_i|}
> \sum_{i=1}^{G}\sum_{t=1}^{|o_i|}
> \min\Big(
> r_{i,t}(\theta)\hat A_{i,t},\operatorname{clip}\big(r_{i,t}(\theta),\,1-\varepsilon_{\mathrm{low}},\,1+\varepsilon_{\mathrm{high}}\big)\hat A_{i,t}\Big]$,
> $\hat A_{i,t}$ can be swapped out with either the GRPO or GDPO advantage estimation.
>
> In fact, our main experiments already adopt the DAPO training recipe (e.g., dynamic sampling, higher clipping, and token-mean loss), as noted in Sec. 4.2. Thus, the “GRPO” and “GDPO” variants in our tables can already be interpreted as DAPO vs. DAPO+GDPO, where the only difference is the advantage estimator (Eq. 2 vs. Eq. 5).
>
> To further validate generalizability, we additionally combine GDPO with both DAPO and CISPO on the tool-calling task, and observe consistent improvements in multi-reward performance under both objectives, as shown in the table above.
>
> | Model | Live Acc ↑ | Multi Turn Acc ↑ | Non_Live Acc ↑ | Correct Format ↑ |
> |-|-|-|-|-|
> | 1.5B-Instruct | 37.89% | 0.12% | 15.63%   | 4.74%   |
> | DAPO| 51.47% | 2.11% | 37.33%  | 75.19%  |
> | DAPO w/ GDPO| 54.39% | 2.35%  | 39.84%  | 81.24% |
> | CISPO  | 46.43% | 1.87% | 35.16%  | 77.48% |
> | CISPO w/ GDPO | 52.41% | 2.08% | 37.72% | 79.15% |
>
> These results show that GDPO is not tied to a specific policy optimization objective, but is a general improvement applicable to a broad class of policy optimization methods. We will clarify this more explicitly in the revised paper.
>
> **Q1**
> ```
> How do weighted rewards affect the performance of GDPO?
> ```
> We already study reward weighting in Sec. 4.2.1 and Appendix K/M, where we vary the length reward weight while fixing the correctness reward weight. Empirically, as shown in Fig. 5, GDPO consistently achieves a better accuracy–length trade-off than GRPO across different weighting configurations. Conceptually, GDPO also provides more direct control over reward trade-offs, since weights are applied after per-reward normalization, making each reward’s contribution to the final advantage more interpretable. In contrast, weighting rewards before joint normalization can distort their intended effect.

---

> > ### Author Rebuttal · Reviewer_faQc · 2026-04-01
> >
> > Thanks for the replies. All my concerns are addressed, and I will update my score.

---

> > > ### Author Response · Authors · 2026-04-01
> > >
> > > Thank you very much for your review and for considering our rebuttal. We appreciate your positive feedback and are pleased to know that our responses have addressed your concerns.

---

### Official Review · Reviewer_CvLK · 2026-03-06

**Soundness:** 2
**Presentation:** 3
**Significance:** 3
**Originality:** 3
**Overall Recommendation:** 5
**Confidence:** 4

**Summary:**

This work addresses reinforcement learning (RL) post-training for large language models (LLMs) in a multi-reward setting.
The authors first show that directly applying GRPO to multiple rewards RL — by summing the rewards for each rollout to obtain the total reward and then normalizing these total rewards within a group—can cause issues such as advantage collapse, loss of signal resolution, and ultimately suboptimal convergence.
To tackle these challenges, they introduce Group reward-Decoupled Normalization Policy Optimization (GDPO).
GDPO first normalizes each reward type within the group, then sums them to create the total reward, and finally performs normalization at the mini-batch level.
Experimental results demonstrate the effectiveness of GDPO across diverse benchmarks, including tool use, mathematical reasoning, and code-based reasoning tasks—showing that GDPO consistently outperforms GRPO in all tested settings.

**Compliance With Llm Reviewing Policy:**

Affirmed.

**Final Justification:**

The rebuttal from the authors addresses my major concerns.

**Key Questions For Authors:**

1. Could you provide the ablation study (as mentioned in W1) and the comparison experiment (as mentioned in W2)?
2. Can this method extend to non-binary rewards (such as continuous rewards)?
3. The batch-wise normalization is normalized over the `data.train_batch_size` (i.e., the batch size of rollout generation) or `ppo_mini_batch_size` (i.e., the batch size of optimization)?
4. Could you provide more discussion on why the second normalization (batch-normalization) is better than the conventional group normalization?
5. Can this work (more specifically, the batch-norm part) apply to the single-reward setting?

**Limitations:**

No discussion of the limitations of this work is provided.

Aside from focusing specifically on the multi-reward setting—which may limit the generality of the approach—I do not perceive any significant additional limitations. However, this specialization does not diminish the value or contribution of the work.

**Strengths And Weaknesses:**

## Strengths

1. Significance: This work is one of the first works targeting the multi-reward setting for post-training, which are increasingly important in LLM post-training.
2. Presentation: This work provides a clear demonstration and analysis on the issues of GRPO for multi-reward settings. The proposed method is clearly presented and explained.
3. Soundness: Even though there is no standalone theoretical support for the proposed method, the empirical justification on GDPO is comprehensive and solid.


## Weaknesses

1. The lack of an ablation study on the 1st normalization -- normalizes rewards independently (Eq. (3)) -- and the 2nd normalization --  (Eq. (5)) batch-wise advantage normalization -- to justify the needs of each component.
2. Lack of comparison against the alternative -- first group normalizes rewards independently (Eq. (3)) and sums the rewards, followed by **group-wise** advantage normalization. I believe this alternative routine is more like a natural generalization of GRPO to multi-reward.


These two weaknesses undermine the soundness of the proposed method, as it remains unclear whether both proposed components are equally important to GDPO.

---

> ### Author Rebuttal · Authors · 2026-03-30
>
> Thanks for your insightful review and feedback.
>
> **W1/W2/Q1/Q4**
> ```
> Could you provide the ablation study (as mentioned in W1) and the comparison experiment (as mentioned in W2)?
> ```
> | Model | Live Overall Acc ↑ | Multi Turn Overall Acc ↑ | Non_Live Overall Acc ↑ | Avg Acc ↑ | Correct Format ↑ |
> |-|-|-|-|-|-|
> | Qwen2.5-1.5B-Instruct| 37.89% | 0.12%| 15.63%| 17.88%| 4.74%|
> | Qwen2.5-1.5B-Instruct-GRPO | 50.63% | 2.04% | 37.87%| 30.18%| 76.33% |
> | Qwen2.5-1.5B-Instruct-GDPO | 55.36%| 2.50% | 40.58% | 32.81% | 80.66%|
> | Qwen2.5-1.5B-Instruct-GDPO w/o Batch Norm | 48.43% | 1.32% | 36.11% | 28.62%| 0%   |
> | Qwen2.5-1.5B-Instruct-GDPO w/ Group Norm  | 50.38% | 1.94% | 37.51% | 29.94% | 74.34% |
>
> We already include evidence in Appendix B showing that removing the batch-wise normalization can lead to unstable training and occasional convergence failure. To further strengthen this point, we additionally ran an ablation on the tool-calling task. We can see that removing the batch-wise normalization causes a substantial degradation in performance, including a collapse of format compliance from 80.66% to 0.00%, and also lowers overall tool-calling accuracy (32.81% → 28.62%). This supports our claim that the second normalization is not cosmetic, but important for stabilizing the numerical scale of the aggregated advantage and enabling stable optimization.
>
> To answer your second question (why can’t we swap out the additional batch-norm with group-norm), The key issue is that applying another group-wise normalization after summing the independently normalized rewards degenerates GDPO back to GRPO.
> Concretely, consider the same toy setting as in Sec. 2 with 2 rollouts and 2 binary rewards. Under GDPO, the reward combination (0,2) yields an advantage of approximately (−1.4142,1.4142)which is distinct from (0,1) and (1,2), both mapping to (−0.7071,0.7071). This separation is what enables GDPO to preserve finer-grained reward distinctions.
> However, if an additional group-wise normalization is applied afterward, (−1.4142,1.4142) is mapped back to (−0.7071,0.7071) making (0,1), (0,2), and (1,2) indistinguishable again. In other words, this extra group normalization re-compresses the reward differences that GDPO is designed to preserve, effectively negating its key advantage. Consistent with this intuition, the “group-norm after reward decoupling” variant performs similarly to GRPO and clearly worse than GDPO in our experiments (see table above). We will include both this intuition and the corresponding empirical comparison in the revised appendix.
> In contrast, as noted in the paper, the batch-wise normalization does not change the number of distinct advantage groups and therefore does not collapse GDPO back to GRPO as the mean and std of batch-norm can be considered as stable constants within group. Instead, it serves to stabilize the numerical scale of the advantages. We are happy to provide additional clarification if helpful. We also hope that the simulation code for Fig. 2, provided in our response to Reviewer z4pa, offers a more concrete illustration of this behavior.
>
> We will make sure to include this discussion in the updated appendix.
>
> **Q2**
> ```
> Can this method extend to non-binary rewards (such as continuous rewards)?
> ```
> Yes. GDPO naturally extends to continuous rewards. In fact, we already evaluate this setting in Sec. 4.3 (coding reasoning), where one of the optimized rewards, the pass rate $R_{\text{pass}} \in [0,1]$, is continuous.
> Empirically, GDPO remains effective in this mixed reward setting. For example, when optimizing both R_{\text{pass}} and the length reward, GDPO improves coding pass rates over GRPO by 3.1% on Codeforces and 3.3% on Taco (Table 3), demonstrating that the method generalizes beyond binary rewards.
>
> **Q3**
> ```
> The batch-wise normalization is normalized over the data.train_batch_size or ppo_mini_batch_size?
> ```
> data.train_batch_size
>
> **Q5**
> ```
> Can this work (more specifically, the batch-norm part) apply to the single-reward setting?
> ```
> Yes, the batch-wise normalization can be directly applied to the single-reward setting. In this case, it reduces to normalizing the advantage values across the batch, which is conceptually similar to standard advantage normalization techniques commonly used in policy optimization [1].
>
> However, we would like to emphasize that this component is not specific to multi-reward settings, nor is it the primary contribution of our work. Its role is to stabilize the numerical scale of the advantages after multi-reward aggregation. In the single-reward case, since there is no reward aggregation step, its effect is typically less critical.
> In contrast, the main benefit of GDPO arises from the decoupled per-reward normalization (Eq. 3), which addresses the reward signal collapse issue that only occurs in multi-reward settings.
>
> [1] What matters in on-policy reinforcement learning? a large-scale empirical study

---

> > ### Author Rebuttal · Reviewer_CvLK · 2026-04-03
> >
> > Thanks for the responses. My concerns are addressed.
> >
> > I will update my score.

---

> > > ### Author Response · Authors · 2026-04-04
> > >
> > > Thank you for your thoughtful review. We appreciate your positive feedback and are glad that our responses helped clarify your concerns.

---

### Decision · Program_Chairs · 2026-04-30

**Decision:**

Accept (regular)

**Comment:**

This paper studies reinforcement learning (RL) post-training for large language models (LLMs) in a multi-reward setting, and argues that directly applying GRPO to aggregated multi-reward signals can collapse distinct reward combinations into identical or overly similar advantages. To address this, the paper proposes Group reward-Decoupled Normalization Policy Optimization (GDPO),  which first normalizes each reward type within the group, then sums them to create the total reward, and finally performs normalization at the mini-batch level. Experiments on tool calling, math reasoning, and coding show consistent improvements over GRPO in both task performance and constraint adherence.
Vanilla GRPO struggles to normalize distinct rollout reward combinations, causing them to collapse into identical advantage values. These identical advantages lead to information loss and poor learning learning. To solve this, the authors introduce Group reward-Decoupled Normalization Policy Optimization (GDPO), which normalizes individual rewards rather than the sum of rewards. This leads to many more configurations that yield a better learning signal. The authors show improvements over GRPO across 5 benchmarks, including AIME and agentic coding.
This work addresses reinforcement learning (RL) post-training for large language models (LLMs) in a multi-reward setting. The authors first show that directly applying GRPO to multiple rewards RL — by summing the rewards for each rollout to obtain the total reward and then normalizing these total rewards within a group—can cause issues such as advantage collapse, loss of signal resolution, and ultimately suboptimal convergence. To tackle these challenges, they introduce Group reward-Decoupled Normalization Policy Optimization (GDPO). GDPO first normalizes each reward type within the group, then sums them to create the total reward, and finally performs normalization at the mini-batch level. Experimental results demonstrate the effectiveness of GDPO across diverse benchmarks.

This is a borderline paper with divergent reviews. The reviewers generally agree that multi-reward RL post-training is an important and clearly motivated problem and appreciate the simple algorithm. There were several concerns on the method and experiments side, which  were largely addressed after rebuttal by adding control baseline and ablations. Reviewers also mentioned that earlier concerns such as  reward weighting and data overlap were clarified. The main remaining concern is originality and significance compared to existing works. In particular, two negative reviewers remain concerned that the core idea is close in spirit to prior decomposition/normalization-based methods such as BNPO, and view GDPO as more of an incremental refinement. At the same time, other reviewers noted that this paper makes a meaningful standalone contribution by analyzing the multi-reward collapse issue in GRPO and proposing a multi-reward post-training solution.

Overall, I lean weak accept. In the final version, the paper should position itself much more explicitly relative to BNPO and other normalization-based approaches, clarifying that the main contribution on the multi-reward diagnosis and discussing if there is any new contribution on the optimization side to decomposition/normalization-based approaches.